# Dimensionality of the reinforced superconductivity in UTe$_2$

L. Zhang [1]✉, C. Guo [1]✉, D. Graf [2], C. Putzke [1], M. M. Bordelon[3], E. D. Bauer [3], S. M. Thomas [3], F. Ronning [3], P. F. S. Rosa [3] & P. J. W. Moll [1]✉

Superconductivity in the heavy-fermion metal UTe$_2$ survives under high magnetic fields, presenting both an intriguing puzzle and an experimental challenge. The non-perturbative influence of the magnetic field complicates the determination of superconducting order parameters in the high-field phases. Here, we report electronic transport anisotropy measurements in precisely aligned microbars in magnetic fields to 45 T. Our results reveal a highly directional vortex pinning force in the field-reinforced phase. The critical current is significantly suppressed for currents only along the **c**-direction, where the flux-flow voltage vanishes with slight angular misalignments—hallmarks of vortex lock-in transitions typically seen in quasi-2D superconductors like cuprates and pnictides. This marks the observation of a transition into a vortex lock-in state at the boundary between two distinct superconducting states. These findings challenge assumptions of nearly isotropic charge transport in UTe$_2$ and point to enhanced two-dimensionality in the high-field state, consistent with a change in the order parameter.

Heavy-fermion superconductivity in UTe$_2$[1–4] presents outstanding challenges to our understanding of strongly correlated materials hosting $f$-electrons. The main open questions evolve around the remarkably rich field-temperature phase diagram that features three distinct superconducting phases (SC1-3)[5–7] including remarkable reentrant behavior at high fields. At zero and low fields, the system shows prototypical heavy-fermion behavior and superconducting properties consistent with a single-component order parameter (SC1)[1,2,8]. For fields close to the crystallographic $b$-direction, a transition into a distinct high-field state (SC2) occurs. Nuclear magnetic resonance and specific heat measurements indicate that this transition is likely accompanied by a change in the order parameter symmetry[9,10]. At higher fields, a uranium-driven metamagnetic transition truncates the superconducting state in a first-order-like transition around 34 T. Lastly, and most enigmatic, is a fully reentrant phase in a small field-angle region around 25°–40° off the $b$- towards the $c$-direction (SC3), whose relation to the other phases remains highly contentious[4,11–13].

These enormous critical fields appear to exceed the Pauli limit[2], which led to proposals of topological odd-parity superconductivity and a surge in experiments investigating the symmetry of superconducting order parameters[8,14–16]. While superconducting states with resilience against high magnetic fields point to unconventional odd-parity states, the tolerance to orbital limiting effects[2] deserves equal attention given its low transition temperature $T_c \sim 2.1$ K.

The first step towards unraveling an unknown superconductor is to establish its phenomenology by determining its Ginzburg-Landau parameters such as the coherence length, which in an orthorhombic system depends on the spatial directions, $\xi_i\{i = a, b, c\}$. Their ratios encode the anisotropy of the superfluid, which is a key parameter to the development of microscopic theories. In addition, reduced dimensionality and fluctuation phase space are key to the physics of cuprates[17], pnictides[18], ruthenates[19], organics[20], and many heavy fermion systems such as CeCoIn$_5$[21], distinct from isotropic unconventional superconductors[22]. The coherence length describes the extent

[1]Max Planck Institute for the Structure and Dynamics of Matter, Hamburg, Germany. [2]National High Magnetic Field Laboratory, Tallahassee, FL, USA. [3]Los Alamos National Laboratory, Los Alamos, NM, USA. ✉e-mail: ling.zhang@mpsd.mpg.de; chunyu.guo@mpsd.mpg.de; philip.moll@mpsd.mpg.de

of the vortex core and its anisotropy sheds light on the vortex core deformation resulting from the superfluid anisotropy.

Conventionally, the direction-dependent Ginzburg-Landau coherence lengths $\xi_i$ are estimated from the orbitally limited upper critical fields along different crystallographic orientations, as $H_{c2}^i = \frac{\Phi_0}{2\pi\xi_j\xi_k}\{i,j,k=a,b,c\}$, where $\Phi_0 = \frac{h}{2e}$ is the magnetic flux quantum. This logic, however, is challenged in UTe$_2$ due to the non-perturbative nature of the magnetic field, which renders the superfluid and its properties strongly field-direction dependent, i.e. $\xi_i(\mathbf{H})$. The dual role of the magnetic field simultaneously inducing orbital effects as well as tuning and transforming the superfluid itself precludes a crisp identification of the superfluid anisotropy. For example, SC2 only exists for magnetic fields applied close to the $b$-direction[23], and critical fields of SC2 along other directions cannot meaningfully be defined or determined. Still, as long as SC2 can be described by a thermodynamic expansion as in Ginzburg-Landau theories, at every point in field a well-defined set of $\xi_i$ and their anisotropies exists. A second complication is the experimental identification of orbitally limiting fields, as other field-tuned effects may suppress superconductivity. SC2 for fields along the $b$-direction is abruptly terminated by a metamagnetic transition well before a putative orbital limit occurs at unknown higher fields. Despite the importance of the field evolution of the superfluid anisotropy in our understanding of UTe$_2$, its investigation is a challenge due to the temperature and field scales of its superconductivity and the non-trivial role of the magnetic field.

This study aims to assess the anisotropy of the high-field phase SC2 by probing the electronic transport anisotropy in the flux-flow regime via critical currents. The anisotropic voltage response for electric currents running in narrow bars along different crystallographic directions allows an estimate of the degree of anisotropy without comparing different field directions. The main finding is a textured, quasi-2D superconducting state in SC2, in contrast to the 3D anisotropic superconductivity at low fields (SC1). The experiments are based on crystalline microstructures of UTe$_2$ carved by focused ion beam[24] from ultraclean crystals grown by the molten salt flux (MSF) method[25].

In the following discussion, it is important to clearly delineate the related concepts of anisotropy and dimensionality. Anisotropy in general describes the dependence of physical responses to excitations in different directions in a crystalline solid, and thus denotes all deviations from spherical symmetry (isotropy). Here, we focus on anisotropies of the electronic system. In the normal state, the ratio of resistivities $\rho_c/\rho_a$ denotes the direction dependence of current flow, which encodes the directional dependence of appropriately averaged Fermi velocities. Likewise, a superconducting crystal may respond quite differently to magnetic fields applied along different directions, which leads to the famous anisotropy of the Ginzburg-Landau parameters of $\xi_i$ and the penetration depth $\lambda_i$, which in certain simple systems are mutually related to the resistivity anisotropy and ultimately, the anisotropy of the effective masses. Their anisotropy describes that magnetic fields penetrate the crystal in substantially different ways depending on their direction.

This concept is distinct from dimensionality, which denotes the degrees of freedom necessary to describe the electronic system. A Fermi surface is classified as two-dimensional (topologically equivalent to a cylinder) if it supports open orbits along one crystallographic direction, a property governing the materials magnetoresistance[26]. A related notion of dimensionality appears in layered superconductors such as certain cuprates, in which the superfluid density is strongly suppressed between highly conductive sheets. When the coherence length in the out-of-plane direction falls below the interlayer distance, the coupling between unit cells changes to a Josephson nature, which has profound implications on their critical phenomena as well as the interlayer vortex matter. The observation of this type of layer decoupling in the SC2 state of UTe$_2$ is the key aspect discussed here.

These concepts are related but not interchangeable. For example, a normal metal with a quasi-2D Fermi surface may still show isotropic resistivities. This happens when a cylindrical Fermi surface is strongly warped such that the Fermi velocity average does not depend much on direction. Equally, a superconductor may be highly anisotropic ($\xi_c \ll \xi_a$) and at the same time clearly three-dimensional, if the out-of-plane coherence length exceeds the interlayer distance.

## Results

Here we investigate the dimensionality and anisotropy of the electron system of UTe$_2$. The resistivity along all three crystalline directions of this orthorhombic compound is measured in carefully aligned microstructures with three different geometries (see the SEM images in Supplementary Fig. 1 and dimensions of all samples in Supplementary Table 1). An "L-bar" geometry is machined from the crystals as shown in Fig. 1a, in which two rectangular bars angled at 90° allow two four-probe measurements in series. Reducing the bar cross-section to the micron-scale is key to achieving the high current densities required to enter the flux-flow regime well below $H_{c2}$, which will provide additional, non-linear insights into transport complementing previous magnetoresistance studies on bulk crystals at low current density[5–7]. Two L-bar microstructures, S1 featuring $a$- and $c$-channels (with cross-sectional area of $S_a^{S1} = 2.54 \times 4.46 = 11.33\,\mu m^2$, $S_c^{S1} = 3.3 \times 4.46 = 14.72\,\mu m^2$, and length of $l_a^{S1} = 22.64\,\mu m$, $l_c^{S1} = 10.69\,\mu m$) and S2 featuring $b$- and $c$-channels (with cross-sectional area of $S_b^{S2} = 3.31 \times 5.46 = 18.07\,\mu m^2$, $S_c^{S2} = 2.82 \times 5.46 = 15.40\,\mu m^2$, and length of $l_b^{S2} = 16.19\,\mu m$, $l_c^{S2} = 10.4\,\mu m$), respectively, have been studied to provide a complete picture of the anisotropy. A straight-bar microstructure S3 with current channel along $c$-axis (with cross-sectional area of $S_c^{S3} = 3.11 \times 4.26 = 13.25\,\mu m^2$ and $l_c^{S3} = 9.46\,\mu m$) was designed for precision field-angle mappings of the flux-flow state.

Focusing first on the normal state resistivity (Fig. 1b, c), $\rho_a$ and $\rho_b$ increase with decreasing temperature until reaching their maximum value at 60 K and 90 K. This typical behavior in Kondo lattices has been associated with the onset of Kondo coherence[27]. The out-of-plane resistivity, $\rho_c$, is markedly different: on cooling, it shows metallic behavior at high temperatures until 60 K, followed by a sharp peak around $T^* = 15.5$ K well below the Kondo scale of $\rho_a$ and $\rho_b$, but is close to the Kondo coherence temperature estimated by STM measurements[14]. This sharp peak in $\rho_c$ has been associated with magnetic fluctuations[27] or a much lower Kondo coherence temperature due to orbital-selective Kondo scattering[28]. At $T_c \sim 2.05$ K (defined by $\rho = 0$), a sharp transition into the noise floor is observed, indicating a robust superconducting state. The sharpness of the transition indicates the absence of strain gradients, whereas subtle device variation of $T_c$ and $H_{c2}$ indicates some homogeneous strain arising from differential thermal contraction (see Supplementary Discussion for strain estimations). Given their similar behavior in temperature, not surprisingly $\rho_a/\rho_b$ is relatively temperature-independent with a rather low anisotropy of 2-3 (Fig. 1c). The largest resistance anisotropy at any temperature is found with respect to $\rho_c$. The ratio $\rho_c/\rho_b$ is ~ 5 at 300 K and remains stable as the temperature decreases until around 20 K, below which it increases rapidly and reaches a factor of 50 just above $T_c$. Overall, the picture of an anisotropic conductor emerges consistent with the current quasi-2D Fermi surface models of corrugated cylinders[29] (see Supplementary Discussion).

These results agree reasonably well with a previous resistivity anisotropy study[27], yet notable differences exist. First, we consistently observe the lowest resistance at all temperatures in the $b$-direction, perpendicular to the uranium chains along the $a$-direction, while ref. 27. observes the lowest resistance along the $a$-direction. As both measurements consistently observe the field-reinforced phase at $H \parallel b$, accidental misalignment can be safely excluded. At room temperature, the resistivity magnitudes in microbar measurements in zero field are in quantitative agreement with the bulk crystal from which

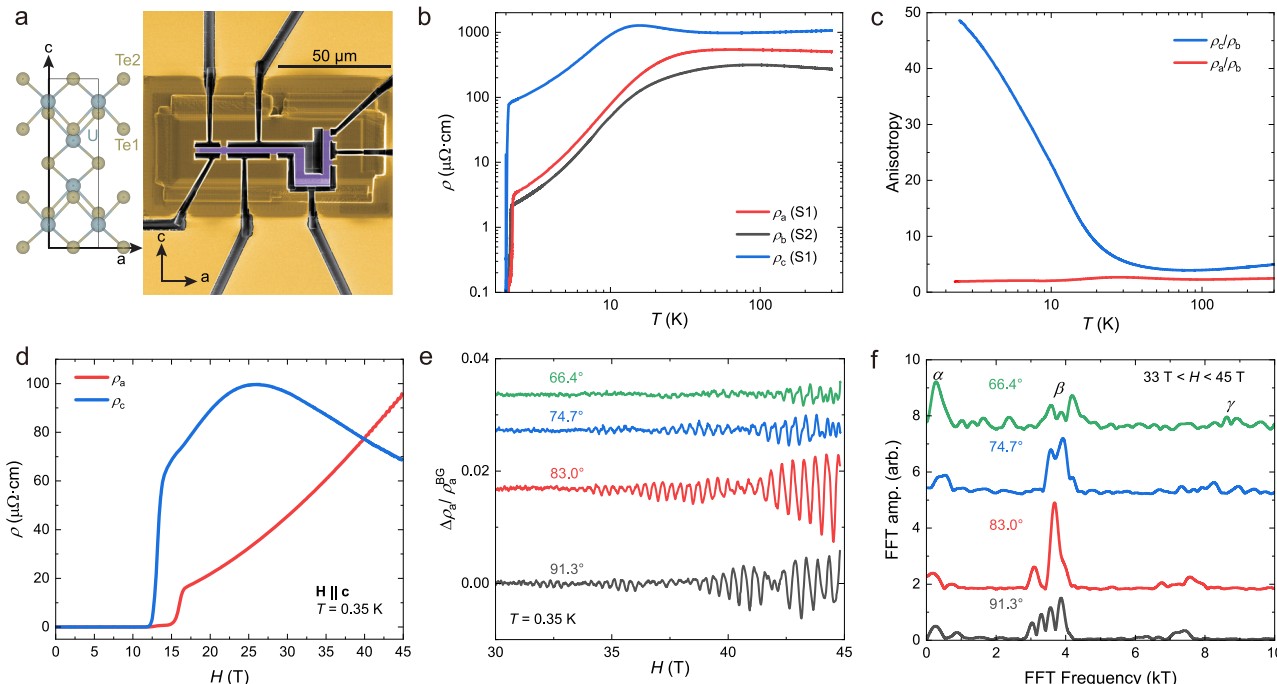

**Fig. 1 | Basic properties of UTe₂. a** The unit cell of UTe₂ (left) and SEM image of a FIB microstructure with channels along the $a$- and $c$-axis (right). **b** The resistivity along different crystallographic axes from 300 K to 2 K. **c** Resistivity anisotropy of UTe₂ from 300 K to 2 K, expressed in $\rho_c/\rho_b$ and $\rho_c/\rho_a$. **d** Magnetoresistivity in 0- 45 T for the magnetic field along the $c$-axis, with the current along the $a$- and $c$-axis at $T = 0.35$ K. **e** Shubnikov-de Haas oscillation when the magnetic field is rotating in $bc$ plane, with the angles between $H$ and $b$-axis labeled. **f** FFT spectrum of the Shubnikov-de Haas oscillations, with a field window from 33 T to 45 T.

they were obtained and exclude potential fabrication-related issues (see Supplementary Fig. 3). The quality of the structures is further evidenced by high residual resistance ratios (RRR = $\rho(300\,\text{K})/\rho(0\,\text{K})$) along $a$- and $b$-axes (RRR$|_{\rho_a} \approx 1000$, RRR$|_{\rho_b} \approx 350$), as well as the sharpness of the transitions and the clear presence of quantum oscillations. Second, ref. 27. observes a remarkably low value for $\rho_c$, which even falls below $\rho_b$ at intermediate temperatures. This led to the key interpretation of the normal state as a rather isotropic conductor, which contradicts electronic structure models that propose only a corrugated cylindrical Fermi surface along the $c$-direction. As a result of this observed isotropy, the presence of a small, closed 3D Fermi surface is proposed. The existence of such a pocket is under active debate, with quantum oscillation evidence for it only observed in tunnel diode oscillator measurements, being challenged by its absence in electrical resistivity and ARPES measurements[30]. These results and recent bulk measurements[31,32] paint a picture of a more anisotropic normal state. A possible route to reconcile these experiments would be a non-trivial role of chemistry given the well-known impact of the synthesis method and defect densities on the correlated state, for example via the formation of defects with direction-dependent scattering.

The magnetoresistance of the normal state shows a non-trivial field dependence from a competition of orbital effects and magnetic tuning of the 5$f$ electrons. As no exotic high-field behavior or reentrance is observed for fields along the $c$-direction, here one may expect a more conventional orbitally limited upper critical field ($H_{c2}$) that may be determined from the magnetoresistance in the $\rho_a$ and $\rho_c$ channels (Fig. 1d). The channels show a difference (2.6 T) of upper critical field of the SC1 phase, consistent with their small difference in $T_c$ likely rooted in the non-zero thermal differential contraction as well. When $H > H_{c2}$, $\rho_a$ shows a $B^2$-dependent increase, typical for the transverse magnetoresistance of a metal. Clear Shubnikov-de Haas (SdH) oscillations appear on top of this background (Fig. 1e, f), providing evidence for the high sample quality and homogeneity in the FIB microstructures. The oscillation frequencies and their angle dispersion are consistent with

previous quantum oscillation results[29,33] supporting a quasi-2D fermiology of UTe₂ (see Supplementary Discussion), and do not show any sign of small 3D Fermi surface pockets. Naturally, this experiment reveals only a few of the lightest orbits, given the relatively high sample temperature in the 45 T magnet ($T \gtrsim 300$ mK) and the heavy masses in UTe₂. In contrast, $\rho_c$ first increases up to a shallow maximum around $H \sim 26$ T, followed by a small decrease. Such behavior is commonly seen in the longitudinal magnetoresistance of magnetic materials and attributed to a suppression of spin-scattering with polarization of moments in the absence of orbital effects.

Overall, a consistent picture of an anisotropic normal state emerges, which reflects in the moderate anisotropy of the low-field superconductivity SC1. First we demonstrate the challenges to capture the anisotropy of field-tuned superconductors from experimental upper critical fields. Zero-temperature critical fields $H_{c2} \parallel a = 8$ T and $H_{c2} \parallel c = 15$ T may be easily obtained as the boundary between SC1 and the paramagnetic metallic state from our data consistent with previous reports[7], yet $H_{c2} \parallel b$ of SC1 is difficult to access as the nature of the SC1-SC2 transition remains to be clarified. One might be tempted to assume $H_{c2} \parallel b = 20$ T, following the zero-temperature extrapolation of the heat capacity anomaly[9] in line with ac susceptibility measurements[10,34]. Under these assumptions, one would estimate the Ginzburg-Landau coherence lengths of SC1 at zero temperature as $\xi_a = 3.3$ nm, $\xi_b = 8.4$ nm, $\xi_c = 6.3$ nm from $H_{c2}^x = \phi_0/2\pi\xi_y\xi_z$. These numbers are clearly unreasonable. The low conductivity direction in the normal state is along the $c$-axis, hence one would expect a low Fermi velocity $v_{F,c}$ and $\xi_c$ to be the shortest. Recent STM studies[35–37] have uncovered vortex cores highly elongated along the $a$-direction, yielding $\xi_a \sim 12 - 15$ nm and $\xi_{[01-1]} \sim 4 - 5$ nm. Importantly, these studies work in the low-field regime of isolated vortices, where field-tuning likely is not yet important. As UTe₂ only cleaves in the (011)-plane, STM cannot obtain $\xi_b$ and $\xi_c$ independently, still it is evident the critical field analysis overestimates them.

The failure of the critical field analysis directly evidences that superconductivity terminates at the SC1 boundary not from orbital

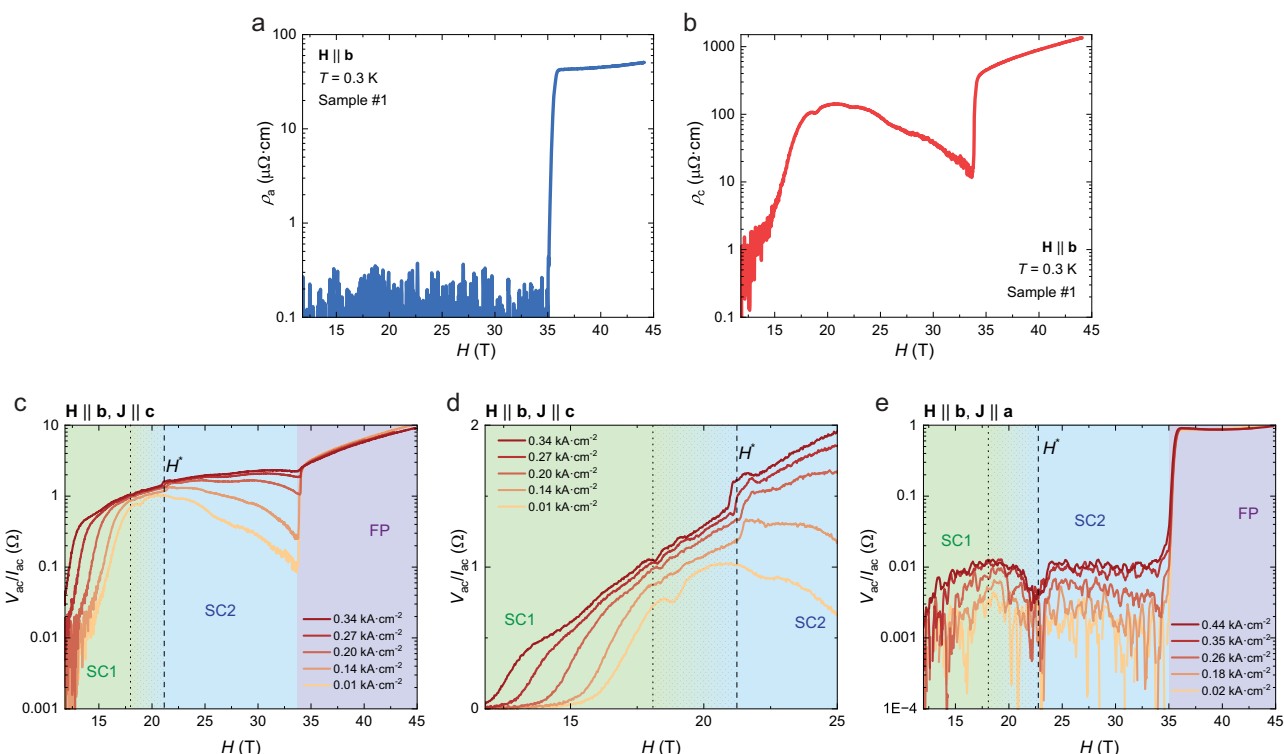

**Fig. 2 | Anisotropic flux flow in SC2 phase of UTe₂.** **a**, **b** Resistivity $\rho_a$ and $\rho_c$ measured at low current densities ($J \parallel a = 0.019$ kA/cm², $J \parallel c = 0.014$ kA/cm²) when $H \parallel b$, $T = 0.3$ K. Nonlinearity of flux flow signal when $H \parallel b$, with $J \parallel c$, on **c** logarithmic and **d** linear scale. **e** Nonlinearity for $J \parallel a$.

effects, consistent with other observations. Already in the rather conventional $H \parallel c$ direction without reentrance, a clear deviation from the Werthamer-Helfand-Hohenberg behavior describing limiting through orbital effects has been observed[2,9]. This directly reflects the difficulty of assessing anisotropy in such field-tuned systems. Importantly for the following vortex lock-in discussion, the Abrikosov vortex cores observed by STM greatly exceed the unit cell size in any direction by a comfortable margin and confirm SC1 as a moderately anisotropic, 3D superconductor.

Next, we examine the field-reinforced SC2 phase for fields precisely aligned with the *b*-direction via flux-flow measurements. In this state, $\rho_a$ and $\rho_c$ measured at low current densities ($J < 0.019$ kA/cm²) display distinct field dependencies (Fig. 2a, b). $\rho_a$ shows zero resistivity up to the metamagnetic transition. Some variance between different samples is observed, $H_m \sim 33.7 - 34.7$ T. This difference might come from the thermal contraction affecting $T_c$ and $H_{c2}$ similarly. Meanwhile, it is consistent with the variance between bulk samples reported in literature[38,39] ranging between 33.1 T and 34.8 T (see Supplementary Table 4), and reflects the still uncontrolled chemical complexity of UTe₂. Note the field zero offset of 11.5 T, the minimal field of the hybrid magnet system at the NHMFL in Tallahassee, below which robust zero resistance has been observed under all conditions in a superconducting magnet (see Supplementary Fig. 4). Given the high vortex density of $1.7 \times 10^4 \mu m^{-2}$ at these fields, the zero-resistance state indicates a relevant region in phase space characterized by well-pinned vortices. In contrast, a large signal in $\rho_c$ emerges at the same current density. The voltage response reaches levels in the same order as the normal state resistance in zero field, 0.1 m$\Omega$·cm, indicating a substantial flow of vortices. This highly anisotropic response to rather low current densities already suggests a significantly anisotropic superconducting state in SC2.

To further investigate the nonlinearity in the flux-flow state (Fig. 2c–e), a direct current (dc) $I_{dc}$ offset was superimposed on a small alternating current (ac) $I_{ac} = 2$ μA (*a*-channel: 0.019 kA/cm², *c*-channel:

0.014 kA/cm², reflecting their slightly different geometric factors) applied to the sample. Self-heating effects were carefully checked for (see Supplementary Discussion). Tilting the field beyond 20° drives the system into the normal paramagnetic phase, which serves as an ideal reference under the exact same thermal conditions and Joule heating power. The dc-currents were limited well within the linear response of the metal.

The strong field dependence of the ac-resistance $R_{ac}^c = V_{ac}/I_{ac}$ for $J \parallel c$ reveals rich vortex physics as the system is tuned by the magnetic field. Starting from a robust zero-resistance state at low fields, flux-flow voltage grows as the boundary between SC1 and SC2 is approached. This initial region is characteristic for an approaching irreversibility line[40], with a gradual softening of the critical currents. This is followed by a narrow region of quasi-ohmic behavior characteristic of free vortex flow as in a Bardeen-Stephen picture[41], which is indicated by a similar ac voltage for all applied biases. Such typical transitions between vortex solids and liquids mark the irreversibility line crossing into a free flow of vortices within SC1. The collapse of the hysteresis of magnetostriction experiments[9] suggests a vortex liquid phase between 15 T and 22 T, which quantitatively matches our transport data (shaded region in Fig. 2c–e). Further signatures for an irreversibility line at this field boundary have been reported in ac-susceptibility and the *a*-direction resistance, $\rho_a$, from which an intermediate vortex liquid phase in-between SC1 and SC2 has been deduced[34,42]. Beyond 22 T, the curves deviate again, indicating the non-ohmic response of finite pinning within SC2. Further increasing the field suppresses the voltage at equal drive. Considering the behavior of $T_c$ being enhanced with increasing field in SC2 phase, a natural explanation for this result is the $\delta T_c$ type of pinning[43], which describes the pinning force induced by disorder in $T_c$ and scales with $(1 - T/T_c)^{-1/2}$. Interestingly, closer inspection of the transition region under finite bias exhibits a sharp step at $H^* \sim 21.7$ T, approximately coinciding with the irreversibility line on the high-field side and delineating the quasi-ohmic from the activated transport region ($H > H^*$). The step marks a clear signature in

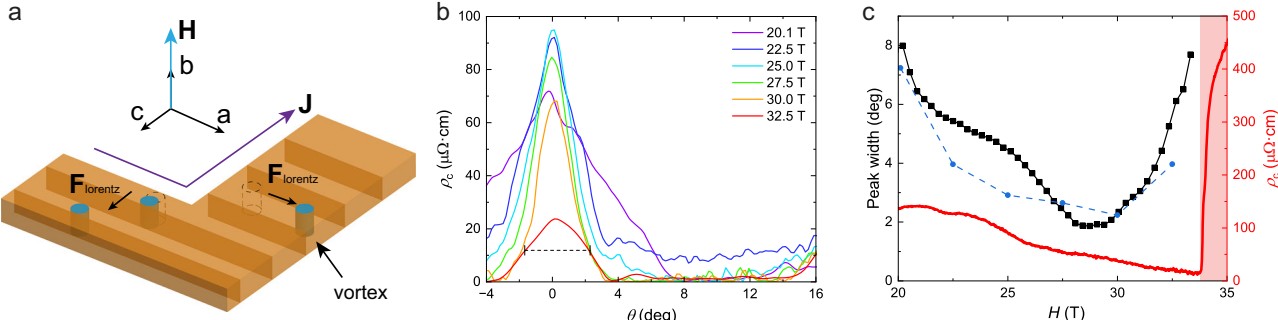

**Fig. 3 | Layered superconducting texture. a** Sketch of quasi-2D model of vortex motion in UTe$_2$. **b** Angular sweep of $\rho_c$ with magnetic field of different strengths rotated from $b$-axis to $c$-axis measured at $J = 0.0076\,\text{kA/cm}^2$, $T = 0.36\,\text{K}$. The dashed line labels the full width at half maximum of the 32.5 T curve. **c** Peak width $w$ (square black dots) of the Lorentzian fit from data points at 0, 4, 8, 9.5, 11.8 degrees and fixed field values from 20 T to 33.5 T, compared with the FWHM (round blue dots) determined from the angular sweep shown in (**b**). The red shaded region labels the field polarized phase above metamagnetic transition.

the vortex response following the SC1-SC2 transition, which happens at lower field in the middle of the vortex liquid phase at $H \approx 19$ T, $T = 0.3$ K according to specific heat measurements[9]. Recent suggestions of an order-parameter symmetry change are compatible with such an instantaneous response in the vortex matter[44]. The occurrence of a discontinuous response may well be a result of the unconventional situation of approaching a vortex irreversibility line from a vortex solid state at high fields, a situation unique to field-driven SC-SC transitions. Indeed, the step-like transition moves to lower fields under increasing bias currents. This is opposite to expectations from self-heating effects, as the irreversibility line on the high-field side has been found to move to higher fields with higher temperatures[9] due to the field-reinforcement of SC2. It will be interesting to explore the natural hypothesis that the irreversibility line is shifted by bias currents towards lower fields.

The situation for current along the $a$-direction, however, is clearly different. $\rho_a$ remains at low-resistive conditions up to the highest applied current densities of $0.44\,\text{kA/cm}^2$, indicative of a well-pinned vortex system. A weak suppression of flux flow resistance is found around 22 T, which correlates with the $H^*$ step in the $c$-direction measurements. Together, this implies an increase in vortex mobility at $H^*$ when driven by $c$-direction currents and suppression when driven by $a$-direction ones, with a substantially lower critical current for the $c$-direction. The observation of suppressed vortex mobility in this field-current configuration at lowest temperatures matches well with experimental reports on macroscopic samples[42].

Changing the field direction provides further insights into the emergence of flux flow dependent on current direction (Fig. 3). The analysis focuses on the region of small angles $\theta$ between the field and the $b$-direction ($\theta = 0°$), rotating towards the $c$-direction ($\theta = 90°$). The enhanced flux-flow resistivity $\rho_c$ is only found for fields well aligned along $b$-direction and sharply decays at small misalignment angles, yielding a sharp peak of $\rho_c(\theta)$. Over most of SC2, the flux flow voltage vanishes within less than $4°$ of misalignment. Sub-degree precision control over field angles is challenging in high magnetic fields due to torque effects on the rotator and limitations of precision measurements of angles via the Hall effect. Thus two independent methods to map the sharp peak are applied. First, the angle is scanned slowly at constant magnetic field strength (Fig. 3b), mapping directly the peak shape. Second, the field is swept at constant angles, providing a more continuous picture in magnetic fields. From these datasets, the full-width at half-maximum is obtained either directly or via a Lorentzian fit for the fixed-angle scans (see Supplementary Discussion).

Both methods consistently uncover a picture of a flux flow occurring only in a sharply confined vicinity of the $b$-direction, and expose a non-monotonic trend in magnetic field (Fig. 3c). Starting out

rather broad around FWHM ~ $7°$ at 20 T, increasing the field narrows the response down to a minimal peak width of $2°$ around 30 T. Further increasing the field broadens the peaks, while the overall flux flow voltage at $H \parallel b$ continues to decrease. This increase is reminiscent of the growth of the quadratic temperature coefficient of resistance in a recent Kadowaki-Woods analysis[38], suggestive of a fluctuation-driven growth of the coherence length and concomitantly a less anisotropic behavior as the metamagnetic transition is approached.

The $a$-axis flux flow reacts to a similar angle range (Fig. 4). While $\rho_a$ maintains at zero-resistance at the low probing currents for fields along $b$ ($\theta = 0°$), dissipative behavior onsets at small angles ($\theta > 4°$). A notable dissipative band emerges around 20 T, reflecting the vortex liquid phase also observed in $\rho_c$. This agrees well with bulk flux-flow experiments of $\rho_a$ slightly off the $b$-direction[42]. Within SC2, $\rho_a$ shows a robust zero-resistance state up to high angles above $10°$, which likely is not due to vortex anisotropy but a result of the weakening order parameter as the system approaches the boundary of the field-reinforced phase upon field rotation. Accordingly, it returns to the normal state value at around $20°$.

## Discussion

The flux-flow dynamics probed by both current channels are sensitively dependent on the vortices' alignment with the $b$-axis of the crystal. Such sharp lock-in effects are commonly observed in the intrinsic pinning regime of layered superconductors such as pnictides, cuprates, or organics[45–47]. This regime occurs when the vortex core diameter in the out-of-plane direction, $\sim 2\xi_c$, is compatible with the interlayer separation[45], $d_c \geq 2\xi_c$, and thus the vortex cores are confined to reside between planes of high superfluid density (Fig. 3a). A highly directional barrier results, with easy vortex motion along the planes, driven by out-of-plane currents, and strong pinning for vortex motion perpendicular to the planes, driven by in-plane currents. A hallmark of intrinsic pinning is its sharp dependence on the field angle. At a small out-of-plane angle, the vortex system realigns with the external field by crossing through the planes, inducing so-called "pancake vortices". The directional pinning is lost at that point, and in particular, the free flow of vortices parallel to the planes stops, as seen here in UTe$_2$. Acceptance angles around $5°$ are typical in structurally layered, highly anisotropic superconductors such as iron-based SCs[45], organic SCs[46], and cuprates[47]. Hence, the observation of sharp intrinsic pinning suggests quasi-2D behavior in UTe$_2$, under the implicit assumption that a $4°$ field rotation does not fundamentally alter the order parameter within SC2. Indeed this is in line with a quasi-2D Fermi surface shown by quantum oscillations and ARPES experiments[29,30,33]. The field-tuned nature of UTe$_2$ is directly reflected in the broadening of the lock-in peak with increasing magnetic field beyond 30 T (Fig. 3c). In conventional systems like

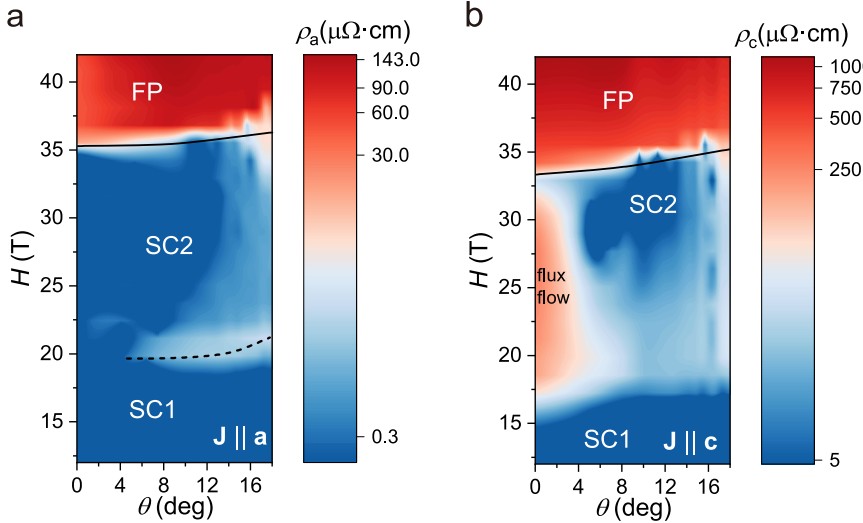

**Fig. 4 | Comparison of current direction dependence on flux flow voltage, for. a** $J \parallel a = 0.019$ kA/cm$^2$, and **b** $J \parallel c = 0.014$ kA/cm$^2$ at $T = 0.3$ K. The dashed line labels the dissipative band appearing in $a$-axis transport around 20 T. The solid line is a guide to the eye for the phase boundary of field polarized (FP) phase.

cuprates, the peak narrows continuously with increasing field[48], owing to the more stringent requirements on alignment perfection between the vortices and the crystal lattice at higher vortex density to avoid a critical number of vortices crossing the planes – thus immobilizing the state.

The experimental discovery of vortex lock-in by itself highlights the unconventional physics of UTe$_2$. To the best of our knowledge, this marks the first observation of a lock-in confinement at a superconductor-superconductor transition. In previous cases, the anisotropy was set by field-independent Ginzburg-Landau parameters and the lock-in state was either entered directly, by applying field to a zero-field-cooled sample, or through a thermal Abrikosov phase by cooling through $T_c$ under finite field[43]. In contrast, a novel route appears in UTe$_2$. Fields up to 20 T induce a dense vortex matter in the 3D anisotropic state of SC1. It is then the phase transition SC1-SC2 which suddenly exposes this vortex matter to a layered potential, causing it to lock-in. Such a situation calls for an entirely different theoretical treatment compared to the usual pathway via an Abrikosov phase.

A natural question follows how UTe$_2$ can support vortex lock-in states in the first place. Although the anisotropic normal state justifies modeling UTe$_2$ as a quasi-2D superconductor, the dense crystal structure makes it difficult to satisfy the defining criterion $d_c \geq 2\xi_c$. The upper bound for a coherence length compatible with intrinsic pinning, $\xi_c = \frac{d_c}{2} = 0.7$ nm, indicates a remarkably small value, similar to that observed in high-$T_c$ materials with massive upper critical fields. If one keeps the low-field value of $\xi_a \sim 14$ nm, the orbital limit along $b$-axis thus coincides with the metamagnetic transition, $H_{c2}^b = \phi_0/2\pi\xi_a\xi_c = 34$ T. This coincidence, however, appears accidental as the transition at $H_m$ clearly is a first-order magnetic one and the order parameter appears to be growing with increasing field, contrary to the expectations if an orbital limit were approached. Thus, a putative orbital limit of SC2 likely occurs at higher field, could the metamagnetic transition be avoided.

The observation of a lock-in state hence implies at least a 5-8 fold reduction of $\xi_c$ at the SC1-SC2 transition, reflecting in an 5-8 fold increase in Ginzburg-Landau anisotropy provided $\xi_a$ and $\xi_b$ remain unchanged. This degree of coherence length anisotropy, $\gamma = \xi_a/\xi_c \approx 20$, would be exotic even considering the normal state anisotropy of 50. As the lock-in effect implies commensurability between the vortex core and the superfluid texture, an alternative scenario would be the formation of a superfluid modulation larger than the unit cell. In such a layered pair-density-wave-like scenario (PDW), $\xi_c \sim \frac{l_c}{2}$, where $l_c$ denotes the periodicity of the superconducting modulation along the $c$-direction. While our experiment cannot distinguish between these scenarios, the PDW case provides a unified picture of the superconducting phases of UTe$_2$, consistent with several experimental observations. The PDW period may be multiple unit cells, thus relaxing the extreme anisotropy scales required by atomic-scale coherence lengths. In a U-dimer system, valence-, spin- or multipolar modulations provide low-energy mechanisms for a material to electronically self-layer. Within that scenario, the high-field phase SC3 may be interpreted as a different allowed PDW wave vector along the (011)-diagonal. Despite the absence of a bulk signature[49,50], an instability towards a PDW already of SC1 has been suggested by STM in zero field[51] as well as evidence for its field-modulation[52].

Overall, our flux-flow experiments on UTe$_2$ have directly observed anisotropy and real-space texture in the superconductivity of SC2, which follows the electronic anisotropy suggested by normal state transport and the known Fermi surface anisotropy. The nature of the superfluid resembles that of a layered quasi-2D superconductor, which should put constraints on microscopic theories of superconductivity in UTe$_2$. The observation of vortex lock-in per se fits well into a picture of a modulated superconducting order parameter, with a one-dimensional modulation along the $c$-direction as seen in structurally layered systems such as La$_{2-x}$Ba$_x$CuO$_4$[53]. Uranium compounds, and even elemental uranium, have been host to a wealth of field-induced density wave phenomena[54,55], and highly resilient forms of superconductivity are found once the effective dimensionality is reduced – a common theme in high-$T_c$ materials.

## Methods

### Crystal synthesis and basic characterization

Single crystals of UTe$_2$ were grown through a molten salt technique using an equimolar mixture of sodium chloride and potassium chloride (NaCl + KCl) as reported previously[25]. The crystallographic structure of our crystals was verified at room temperature by a Bruker D8 Venture single-crystal x-ray diffractometer equipped with Mo K-$\alpha$ radiation. To ensure that the samples only show a single superconducting transition temperature, specific heat measurements were performed using a Quantum Design calorimeter that utilizes a quasi-adiabatic thermal relaxation technique.

## Microstructure fabrication

The microstructures of $UTe_2$ are fabricated by Thermofisher Helios Ga FIB or Hydra. The lamellae of the five samples used were cut from an MSF-grown crystal with a superconducting $T_c = 2.1$ K, and then transferred onto a sapphire substrate with Pt welding deposited by FIB. These lamellae were etched with radio frequency (RF) argon plasma to remove surface oxide layer, which was followed by a high-power Au sputtering in the same vacuum chamber to form electric contact. The gold sputtered lamellae were patterned to different geometries and protected with FIB-deposited carbon in Hydra system. The devices were calibrated in Quantum Design PPMS system with 9 T superconducting magnet for the temperature dependence of resistivity.

## High-field measurements

High-field transport measurement was performed in the 45 T hybrid magnet at the National High Magnetic Field Laboratory. Transport signals were read out via Stanford Research 86x lock-in amplifiers. AC and DC currents are applied via the Stanford Research CS580 voltage-controlled current source.

## Data availability

Data that support the findings of this study are deposited to Zenodo with the access link: https://doi.org/10.5281/zenodo.17250203.

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

## Acknowledgements

This work was supported by the DOE Office of Basic Energy Sciences, Materials Sciences and Engineering Division project 'Quantum fluctuations in narrow band systems'. C.G. acknowledges financial support by the European Research Council (ERC) under grant Free-Kagome (Grant Agreement No. 101164280). The high field experiments were performed at the National High Magnetic Field Laboratory, which is supported by National Science Foundation Cooperative Agreement No. DMR-2128556 and the State of Florida.

## Author contributions

Crystals were synthesized and characterized by M.M.B., E.D.B., S.M.T., F.R., P.F.S.R. The experiment was designed by L.Z., C.G., P.J.W.M. The FIB microstructures were fabricated by L.Z., C.G., C.P. The magnetotransport measurements were performed by L.Z., C.G., D.G. The analysis of experimental results has been done by L.Z., C.G., P.J.W.M. All authors were involved in writing the paper.

## Funding

## Competing interests

The authors declare no competing interests.
