## [Transparent Peer Review file · Nature Communications]

Dimensionality of the reinforced superconductivity in UTe₂

Corresponding Author: Mr Ling Zhang

Version 0:

Reviewer comments:

Reviewer #1

(Remarks to the Author)

The authors report electrical resistivity measurements on FIB-microstructured UTe₂ samples and discuss the possible observation of a vortex lock-in effect. Based on these observations, they consider the possibility of quasi-two-dimensional superconductivity in this compound. However, I have serious concerns regarding both the data and its interpretation, as detailed below. Their interpretation relies on data that are questionable. Studies on microfabricated samples must first demonstrate consistency with bulk properties; otherwise, it remains unclear whether microfabrication may have altered the intrinsic behavior of the material. Due to these concerns regarding data quality and interpretation, I cannot recommend publication in Nature Communications.

The key data set is presented in Fig. 4b, which shows vortex flow occurring only near the H // b orientation. The authors interpret this as evidence for vortex lock-in and as support for quasi-two-dimensional superconductivity. Their data suggest that superconductivity (SC2 phase) persists up to at least 16 degrees away from the b-axis towards the a-axis. However, according to Z. Wu et al. (PNAS 121 (37) e2403067121 (2024)), superconductivity is suppressed above 8 degrees away from the b-axis towards the a-axis. This significant discrepancy with bulk results raises serious concerns about the reliability of the present measurements or the quality of the samples.

Furthermore, the metamagnetic transition field reported in Fig. 4b is approximately 33 T, which is clearly smaller than the widely reported value of 34.5 T for H // b. Since 34.5 T represents the lowest metamagnetic transition field for any field orientation, it is highly questionable that the transition field in their measurements appears even smaller, suggesting additional uncertainties.

There have been previous reports on vortex flow in UTe₂. Including the present work, I find three papers addressing flux flow. H. Sakai et al. (PRL 130, 196002 (2023)) reported flux flow across the entire SC2 phase for J // a, while Y. Tokiwa et al. (PRB 108, 144502 (2023)) observed flux flow only in the intermediate field region of SC2 for J // a. In contrast, the present work claims no flux flow at all for J // a. The authors should explicitly discuss the advantages or improvements of their measurements relative to these previous studies. Otherwise, readers are left uncertain as to which result reflects the intrinsic behavior. One important factor is the precision of the current alignment along the a- or c-axis. Ideally, the distance between voltage contacts should be much larger than the sample's cross-sectional dimensions. It remains unclear whether the sample geometry in the present study is superior to that of the previous work.

I also find the fitting results presented in Fig. 3b unreliable. The data are extremely sparse for fitting with any function: there are not even two data points within the half-maximum width. Furthermore, the authors do not present data for negative field angles, making it impossible to verify that "0°" truly corresponds to H // b. If the actual H // b orientation is slightly offset from 0°, the fitting results would change substantially.

The authors assign the H* anomaly to the phase boundary between SC1 and SC2. However, it is difficult to distinguish this assignment from an accidental coincidence. The authors need to present the temperature dependence of H*(T) and demonstrate that it consistently tracks the SC1–SC2 phase boundary to justify this interpretation.

Finally, in Fig. 2c, the authors report flux flow even when the DC current is zero. I am concerned that this flux flow may be induced by the finite AC current, and it is necessary to verify whether the voltage drops in the limit of zero AC current. The authors should also examine the possible effect of Joule heating. Nearly ohmic behavior is observed around 18–20 T, which coincides with the field region where T_c(B) is lowest. This near-ohmic behavior could result from Joule heating elevating

T_{sample} above T_c specifically in this field range, or simply from a reduced critical current because the measurement temperature becomes closest to T_c at these fields.

(Remarks on code availability)

Reviewer #2

(Remarks to the Author)

Report on the paper “ Electronic dimensionality of UTe2 “ by Zhang et al.

In this work the authors report experimental measurements of the non-linear electrical transport for fields applied around the b-axis in the different superconducting phases of UTe2 in FIB microstructured high-quality (molten flux grown) single crystals. This microstructuring allows to reach very large current densities usually only possible in thin films, and hence to directly study the different flux-flow, flux-creep regimes across the superconducting phase diagram. Moreover, the microstructuring also permits to study the transport for different current directions on the same crystal, which proves to be highly pertinent, notably in the superconducting phase.

However, a first surprising result is the large resistivity observed on high quality samples for current along the c-axis: it was known from measurements on millimeter-size samples that resistivity is larger for current along c than for current along a and b. However, the anisotropy between the c-axis and the other directions was more than 5 times smaller. Everything happens as if the improvement of the RRR on the molten flux sample studied here did not affect the c-axis direction: the values observed here are comparable to those on works a bit more recent than that of Eo et al., performed on CVT samples when the current direction is indeed along the c-axis (see PRB 109, 155103 (2024)). Otherwise, the value of the c-axis conductivity can be significantly smaller (see again PRB 109, 155103 (2024)). Hence, rather than a difference in the anisotropy between CVT-grown and flux grown samples, due to the “non-trivial role of chemistry”, don't these results rather suggest the existence of defects or impurities affecting transport along c-axis more strongly than the sample stoichiometry improved by the flux-growth process?

So, from the observed very large anisotropy of the electrical transport between the c-axis and the a-b direction in these high quality microstructured samples, the authors conclude that UTe2 would be a quasi 2D layered conductor, comparable to the High- T_c cuprates, iron-pnictides or organic systems. This point is indeed a recurrent question in the analysis of the physical properties of UTe2, and the new results presented here on microstructured samples which are both “high quality samples and geometrically well-defined is an important element.

The conclusion is however maybe still not so well firmly established, because if the point was only that of a quasi-2D anisotropy, resistivities should scale in all direction with the RRR. Moreover, as remarked by the authors, this very strong anisotropy of the Fermi velocities would be expected to reflect in the superconducting properties, and notably in the anisotropy of H_{c2} : the author do show that even with the uncertainties arising from the “field-tuned” superconducting properties, and the departure of the temperature dependence of H_{c2} from the conventional WHH predictions, the coherence length along the c-axis far exceeds a putative “interlayer distance”, contradicting the hypothesis of a layered superconductor.

Yet, the core of the present work is certainly the study of the “flux-flow” state in UTe2 for field along the b-axis, both in the low field SC1 phase and in the SC2 phase up to the metamagnetic transition at around 35 T. Several striking results are presented:

- transport along a-axis displays very little flux-flow voltage, showing that vortices remain strongly pinned for current along the a-axis (and field along the b-axis) both in the SC1 and SC2 phases.
- transport along the c-axis displays sizeable “flux-flow” voltage at low temperature above 15 T, and a marked anomaly at the transition between the SC1 and SC2 phases (between 21-22T).

The huge difference between a and c-axis transport is a very new result, and add significant input compared to previous work (ac susceptibility and magnetostriction) which did show the existence of weak pinning areas in the superconducting phase diagram for H//b. There is a good qualitative agreement between the different measurements, and the present work could also study the angular dependence of the flux flow resistivity which typically vanishes above 4° misalignment in the (b,c) plane.

If there is little doubt on this sensitivity to angle, the attempt to fit the angular dependence by Lorentzian forms is surprising, owing to the sparse number of points notably in the region of the line width. Hence, trying to extract quantitative information on the line width as present in Fig 3.c is not convincing and maybe not so important either for the discussion.

Indeed, the main point is clearly the origin of the anisotropy between flux-flow along a and along the b-axis. The authors discuss and exclude a lock-in of the vortices between superconducting planes for the reasons on the coherence length along the c-axis mentioned above. This is perfectly sound, however, the estimation of the lower bound of $H_{c2} = 145$ T in the SC2 phase is difficult to understand: as mentioned before, UTe2 is a “field tuned system” hence it is complicated to understand how to extrapolate H_{c2}/b in the SC2 phase from coherence length in the SC1 phase which depend on the T_{sc} value in that phase. More deeply, what is the meaning of extrapolating H_{c2} in a field-tuned system, and what for?

The second proposal in the discussion is a modulated phase in the superconducting state (“pair-density wave” state). Such a state could indeed relax the constraint on the coherence length along the c-axis having to be smaller than in the interlayer distance. However, the author should clarify why they expect a change of anisotropy between the SC1 and SC2 phases: in their measurement, flux-flow voltage appears also in the SC1 phase and again only on the c-axis transport. Moreover, thermodynamic measurements of H_{c2} have consistently shown that the initial slope of H_{c2} at T_{sc} along the a-axis is larger

than along the c-axis: field-tuning prevents again clear-cut conclusions, but the bare measurements do suggest that coherence length along the c-axis is also the shortest in the SC1 phase, which is in line with the present measurements?

A last point which should be clarified in the discussion is that on the possible Lebed mechanism. Again, this Lebed mechanism seems only possible for reasonable field values only in strongly layered systems, excluded by the authors. Also, the observation of a halo of superconductivity for the field-reentrant phase above H_m seems incompatible with a Lebed mechanism requiring alignment of the field within the layers. Again, this point is confusing and does not seem key for the work.

As a conclusion, I do recommend publication of this work in Nature communication, as it bring new important results on the behaviour of the superconducting phases, with unique measurements on high quality crystals, and suggests an appealing hypothesis of a modulated phase at high-field along the b-axis. As mentioned above, some points should be clarified in the discussion and some additional minor points could also improve the manuscript:

- on line 222, add references for the previous estimates of the coherence length in the a and c directions.
- If available, a complete I-V curve at some field and temperatures would also help distinguish flux-flow- flux creep regimes and appreciate the values of the critical current
- Measurement of H_{c2} in the different directions on this sample would also be helpful
- On figures 2, 3 and 4, it appears that H_m is slightly above 35T on the transport measurements with current along a-axis, and around 33.5T (?) for current along c-axis.

Could the authors comment/explain this discrepancy?

(Remarks on code availability)

Version 1:

Reviewer comments:

Reviewer #1

(Remarks to the Author)

Unfortunately, the authors' reply does not provide a satisfactory resolution to my concerns, and therefore I still cannot recommend this manuscript for publication in Nature Communications. I appreciate the clarification regarding the rotation direction, Joule heating, and the issues in Fig. 3. However, I remain unconvinced by their conclusions, which may largely reflect sample-dependent effects.

I am grateful for the authors' discussion of the flux-flow results in relation to previous studies. Nevertheless, I remain skeptical. The angular mismatch of the previous report is 2° , but according to Fig. 4(a), flux flow does not emerge at such a small deviation; at least 4° appears necessary. Furthermore, earlier reports observed flux flow above 15 T, whereas in this work it only appears above 20 T. The magnitude of the finite resistance also seems very small. Under these circumstances, the claim of consistency with earlier studies is difficult to accept.

I previously emphasized that for measurements with current along the a axis, the ideal sample should be needle-shaped along the a direction. This point was meant to highlight concerns about current redistribution, rather than being a matter of contact configuration. Contact geometry is instead more relevant to Joule heating. The Supplementary Material presents the sample geometries, and among them, S1 ($J \parallel a$) appears to be the most elongated along the current direction. Nevertheless, even in this case ($2.5 \times 4.5 \times 23 \mu\text{m}^3$), the geometry is no better than in an earlier report ($0.73 \times 0.75 \times 4.6 \text{ mm}^3$).

As Reviewer #2 also notes, the c -axis resistivity may strongly depend on sample conditions. Since UTe_2 crystals are known to grow elongated along the a axis while being fragile and prone to exfoliation along other directions, the reliability of c -axis measurements is even more questionable. Discrepancies with prior reports are already evident in the a -axis data, and the origin of these differences remains unclear. It is possible that they arise simply from sample dependence. Therefore, the c -axis results cannot be accepted at face value as intrinsic.

A further point, albeit minor, concerns the definition of H_m in the Supplementary Material. The authors list values from various reports, but these are defined at the onset of the resistivity increase. The proper definition should be at the midpoint. For instance, the reference to Knafo et al. (2021) is cited as yielding the lowest H_m value, yet the original text clearly states $H_m = 34 \text{ T}$. While some works report 34 T and others 35 T, the majority of studies that provide values with decimal precision converge around 34.5 T. Thus, I remain convinced that the value below 34 T reported here is too low. Moreover, in Fig. 4(a) and 4(b), the metamagnetic transition field at zero angle should coincide, but they do not. This inconsistency suggests spatial variations of the physical properties within the sample, further raising doubts about the data quality. Furthermore, in the supplementary material their H_m value for J/a is 34.79T, but Fig. 4a clearly shows H_m above 35T.]

Finally, regarding Fig. 2c, my question (not a critique) was not about the trivial fact that the ac-voltage must vanish identically

at zero ac-current. Rather, the essential point is to distinguish whether the finite resistance observed at zero dc bias is induced by the finite amplitude of the ac excitation current used in the measurement. To clarify this, one would need to plot V_{AC} vs I_{AC} . The value of I_{AC} at which V_{AC} drops to zero corresponds to the critical I_{AC} . I was simply wondering how small this critical I_{AC} is.

In summary, I had asked the authors to demonstrate a clear advantage or improvement of their experiment over previous studies. Unfortunately, no such advantage has been established in the present version of the manuscript.

(Remarks on code availability)

Reviewer #2

(Remarks to the Author)

In this resubmission of their work on "Electronic dimensionality of UTe_2 ", the authors have made their answers to the referees' remarks and added some new results. Reviewing the answers to the referees:

- On the question of a difference between measurements on bulk samples and FIB microfabricated samples: the authors do explain convincingly that there is in fact no discrepancy regarding the results on flux-flow, and that their sample quality and geometrical definition are at least as good as on the bulk measurements. Regarding the value of the metamagnetic field, a similar difference from sample to sample is found in both cases. It is more surprising to find such a variation in the FIB microfabricated samples, because they are cut in the very same crystal and probably in very closed area of the sample. This remains one of the puzzles of this compound but indeed, it does not question the quality of FIB shaped samples.
- Regarding the question of the comparison between the resistivity measurements by Eo et al. on c-axis transport on (early) bulk CVT crystals with the author's results, I understand the point that the paper by Eo has long been central in the discussion of the transport anisotropy in UTe_2 . However, it is also known presently that these difficult measurements were probably done with no perfect alignment of the current with c-axis and so significantly underestimated the c-axis resistivity. Hence, I find that discussing the more recent bulk results would have been helpful to define the puzzle of the c-axis transport and its non-evolution with sample quality.
- Regarding the interpretation of the large anisotropy observed between a,b electrical transport with c electrical transport in the present measurements, I still find the discussion confusing: in their rebuttal letter, the authors, if I understand well, distinguish a " ", which would be realised in the SC2 phase of UTe_2 as demonstrated by the lock-in transition they observed in the vortex state, from a "quasi 2D conductor", which would imply, like for the cuprates, that the coherence length along the c-axis is shorter than the interlayer distance.

However, in the presentation of the results in the main paper (lines 98-99), they state from their transport measurement that UTe_2 is a layered conductor "not quite in the quasi 2D-limit". Maybe the problem is to define what is a layered conductor then? However, from the crystal structure or the cleaving properties, it is not easy to picture UTe_2 as layered (along the c-axis), as opposed to organic, cuprates, van der Waals compounds... An additional confusion comes from the comparison to the Fermi surface (warped cylinders), which the authors claim to be in agreement with their transport measurements: the sentence 114-117 state that: "Our results paint the picture of a much more anisotropic normal state in the MSF crystals, which is further supported by the observation of strong vortex lock-in in the high field phase discussed below. The large transport anisotropy observed here is well explained by a corrugated cylinder alone, suggesting that UTe_2 is a low-dimensional metal".

These two sentences suggest again that UTe_2 is a 2D metal, contradicting observations mentioned above on the SC1 phase, or the global conclusion of the paper that anisotropy in the SC2 phase might arise from a modulation of the superconducting state, not from normal state properties. And makes it hard to understand the reason for the suggestion of anisotropic diffusion on line 120, if the c-axis transport is well explained by the sole Fermi surface?

I would rather agree with this conclusion of anisotropic scattering and with the related discussion in the rebuttal letter, precisely because c-axis transport cannot be explained solely by Fermi velocity anisotropies. Indeed, the author also show (in the SI) that the geometry of the Fermi surface gives an anisotropy between a, b and c axis in the same order as measured by electrical transport. But quantitatively, a factor 3 is missing at the crude level of their calculation, and making more elaborate estimates is beyond reach due to the correlation effects. They also discuss how the coherence length as measured by STM or the anisotropy of H_{c2} in the SC1 phase does not support "layered 2D superconductivity" but rather an anisotropic 3D conductivity.

I do not believe that there is a problem with the conclusion of the authors, but rather a problem in the presentation of the discussion: an external reader will be stunned by the factor 50 anisotropy of the transport, and it would be very helpful to distinguish between what everybody agrees on (that UTe_2 is an anisotropic 3D conductor and superconductor in the low-field phase, with a quasi 2D warped Fermi surface leading to the 3D behaviour, with what is (according to the authors) really supporting/explaining/or contradicting the new result on the factor 50 anisotropy between ab and c resistivities.

Regarding also the question of anisotropy of the coherence length, directly proportional to the Fermi velocities, the authors still present, with caution, their estimation from extrapolation of the upper critical field to zero temperature, explaining that in a field-tuned system, they cannot be relied-on. In this new version, they added references to the recent STM work on the vortex lattice. Let us remark that these STM estimates of the coherence length along the different axes (directly proportional to the average Fermi velocity in that direction) are in very good agreement with previous estimations deduced from H_{c2} measurements, taking account of the field tuning and using the initial slope of the upper critical field (at T_c) rather from the zero-temperature extrapolation (see references 9 and 33).

- On the main result of the paper, namely the very large anisotropy in the SC2 phase between flux-flow resistance for current along the c or along the a-axis, the authors have clarified several points as explained in their rebuttal letter, and their conclusion appears more clearly.

- On point raised by both referees was the lack of points to discuss the shape of the angular dependence of the lock-in transition. This was corrected in this new version thanks to new measurements with continuous rotation. The discussion is also clear except for the last sentence (lines 224-226): "This increase is reminiscent of the growth of the quadratic temperature coefficient of resistance in a recent Kadowaki-Woods analysis³⁵, suggestive of a fluctuation-driven weakening of the superconducting coherence as the metamagnetic transition is approached." This suggests that superconductivity would become weaker due to the increase of thermal fluctuations? However, such an effect is characteristic of the strong coupling regime, and at the opposite, the increase of the A coefficient or of the Sommerfeld coefficient was taken in the community as showing an increase of the pairing strength, quite contradictory with the picture of coherence weakening?

- Regarding the comparison between the thermodynamic phase diagram and that deduced from flux-flow resistance, the authors are cautious in trying to identify anomalies on transport reflecting the SC1-SC2 transition. Nevertheless, or maybe because of this caution, the statements remain ambiguous: lines 190-193 suggest that the step-like accident at 21.7T could mark the SC1->SC2 transition, but lines 195-200 seem to discuss the irreversibility line observed for example on magnetostriction within the SC2 phase, whereas line 200-201 comes back to the SC1-SC2 transition. This is all the more confusing that the SC1-SC2 line is the only one observed at low temperature (.3K), and that it does decrease in field on increasing temperature, which would be compatible with self-heating effects, whereas indeed the irreversibility line has the opposite behaviour, and was not observed at such a low temperature.

- Last, the introduction of the discussion is still confusing: it seems to start with a claim that UTe₂ is a low dimensional quasi2D superconductor finishing line 254, until this starts to be questioned on line 275. Again, there is no problem with the physics, but it could be easier to follow if it was announced earlier that the most straightforward interpretation raises some difficulties requiring to look for another scenario.

As a conclusion, I do recommend publication of this work, which brings new result notably on the nature of the high field superconducting phase. I would just recommend to still try improving readability in clarifying the presentation of the point mentioned above.

(Remarks on code availability)

Response to Reviewers v2

Reviewer #1 (Remarks to the Author):

Unfortunately, the authors' reply does not provide a satisfactory resolution to my concerns, and therefore I still cannot recommend this manuscript for publication in Nature Communications. I appreciate the clarification regarding the rotation direction, Joule heating, and the issues in Fig. 3. However, I remain unconvinced by their conclusions, which may largely reflect sample-dependent effects.

Dear reviewer,

From the comments, a fundamental distrust in micromachining of crystals becomes apparent, which appears robust against experimental evidence. It is difficult to rationalize, on a very fundamental level, why our microbar data supposedly reflects an altered material while we demonstrate clear out quantum oscillations in perfect agreement with dHvA studies, which none of the other transport experiments on bulk crystals were able to do. In our view this represents the gold standard of crystalline quality, especially in the context of heavy fermions.

We share and welcome the critical comments as they help to refine the paper and clarify all technical details.

I am grateful for the authors' discussion of the flux-flow results in relation to previous studies. Nevertheless, I remain skeptical. The angular mismatch of the previous report is 2° , but according to Fig. 4(a), flux flow does not emerge at such a small deviation; at least 4° appears necessary. Furthermore, earlier reports observed flux flow above 15 T, whereas in this work it only appears above 20 T. The magnitude of the finite resistance also seems very small. Under these circumstances, the claim of consistency with earlier studies is difficult to accept.

The first point concerns the angle mismatch, which within condensed matter research must be considered minute. Single-axis rotators used in standard solenoids with manual sample mounting rarely perform much better. Instead, it is gratifying to see how the rough estimate by [Tokiwa, Y. *et al. Phys. Rev. B* **108**, 144502 (2023)] agrees without data. Note that the critical angles of any superconducting feature in it measured on bulk crystals vary by at least 0.7° . The second point concerns the onset field of this flux flow resistance. As shown in the figure below comparing their result to ours at 4° , we find perfect agreement in the onset field.

I previously emphasized that for measurements with current along the a axis, the ideal sample should be needle-shaped along the a direction. This point was meant to highlight concerns about current redistribution, rather than being a matter of contact configuration. Contact geometry is instead more relevant to Joule heating. The Supplementary Material presents the sample geometries, and among them, S1 ($J \parallel a$) appears to be the most elongated along the current direction. Nevertheless, even in this case ($2.5 \times 4.5 \times 23 \mu\text{m}^3$), the geometry is no better than in an earlier report ($0.73 \times 0.75 \times 4.6 \text{ mm}^3$).

We respectfully disagree with the reviewer's statement on the irrelevance of the contacts with regard to the current redistribution; this is a key component to any critical current measurement. The contacts formed by colloidal silver are prone to inhomogeneous current injection, which is why in an applied physics setting critical currents are always measured with soldered contacts. Thus the shape and local details of the silver paint contacts are most crucial. We further disagree that an elongated needle-shaped sample would be ideal for critical current measurements, the opposite is true. Critical current measurements are performed best in constriction geometries, with well-defined minima in the cross-section that serve as short, defined regions of maximal current density. It is important to have large contact areas, to avoid heat accumulation dominating a long needle as well as the key issue of quenching at the contacts and current injection points. Our samples reflect this, with large parts of the crystal allowing low-dissipation current injection while having a narrow cross-section between the voltage probes.

As Reviewer #2 also notes, the c -axis resistivity may strongly depend on sample conditions. Since UTe_2 crystals are known to grow elongated along the a axis while being fragile and prone to exfoliation along other directions, the reliability of c -axis measurements is even more questionable. Discrepancies with prior reports are already evident in the a -axis data, and the origin of these differences remains unclear. It is possible that they arise simply from sample dependence. Therefore, the c -axis results cannot be accepted at face value as intrinsic.

We would like to point out that reviewer #2, instead, claims that so far the disagreement between different bulk results comes from mostly current alignment. Regarding the differences between bulk and microstructures of layered materials, the exact opposite is true. Microstructures allow a direct inspection of the material probed, whereas a bulk crystal measurement averages over largely unknown material.

We agree with the reviewer's comment that the reliability of c -axis measurement is affected by the crystal growth directions, but this is exactly one of the reasons that FIB microstructuring is much more reliable in measuring anisotropy than bulk measurement. All the micro cracks, exfoliations, twin domains that might affect crystallinity of bulk samples are substantially reduced in a FIB microbar, as it allows direct and microscopic inspection and analytics of the material under study – unlike the bulk of a macroscopic crystal. Based on the reviewer's concern on sample conditions, we claim that our c -axis resistivity study is the most trustworthy result so far compared to other bulk studies, as reflected by the comments of reviewer #2.

A further point, albeit minor, concerns the definition of H_m in the Supplementary Material. The authors list values from various reports, but these are defined at the onset of the resistivity increase. The proper definition should be at the midpoint. For instance, the reference to Knafo et al. (2021) is cited as yielding the lowest H_m value, yet the original text clearly states $H_m = 34 \text{ T}$. While some works report 34 T and others 35 T, the majority of studies that provide values with decimal precision converge around 34.5 T. Thus, I remain convinced that the value below 34 T reported here is too low. Moreover, in Fig. 4(a) and 4(b), the metamagnetic transition field at zero angle should coincide, but they do not. This inconsistency suggests spatial variations of the physical properties

within the sample, further raising doubts about the data quality. Furthermore, in the supplementary material their H_m value for $J//a$ is 34.79T, but Fig. 4a clearly shows H_m above 35T.

For any transition in a real sample, there is a finite transition width and one could have multiple ways to define the transition, just as common resistance criteria in upper critical field studies one could determine. As long as the arbitrary definition is used consistently, it allows to compare samples. Given the chemical challenges of this compound and the current state of the field, the value of discussing second decimal points of critical fields appears questionable. As the referee states, this minor point is not critical to the current theoretical developments aimed at capturing the qualitative phenomenology of the material, we are far from a microscopic understanding that could refine models based on %-level deviations of critical fields. On the contrary, qualitative results such as the here reported lock-in transition are now needed to constrain such models.

Finally, regarding Fig. 2c, my question (not a critique) was not about the trivial fact that the ac-voltage must vanish identically at zero ac-current. Rather, the essential point is to distinguish whether the finite resistance observed at zero dc bias is induced by the finite amplitude of the ac excitation current used in the measurement. To clarify this, one would need to plot V_{AC} vs I_{AC} . The value of I_{AC} at which V_{AC} drops to zero corresponds to the critical I_{AC} . I was simply wondering how small this critical I_{AC} is.

We apologize for misunderstanding the question. This is an interesting point we could not experimentally solve due to the small signals V_{AC} in the low current limit I_{AC} in the hybrid magnet. Hence we applied the smallest I_{AC} with reliable signal-to-noise ratios, at which there always was a finite resistance. This touches on a crucial point made by the reviewer about the “superiority/inferiority” of our devices compared to macroscopic bulk crystals. In our view, each approach or method has its own parameter space of excellence. Our devices are made to exhibit micron-sized constrictions of the current paths, achieving very high current densities at small overall applied current to minimize self-heating and contact non-linearities. They are excellent to study high bias physics, vortex flux flow etc. Under the special circumstances of vortex lock-in, however, bulk pinning vanishes and only relatively weak surface pinning remains – in effect the critical current density becomes very small. It should indeed be finite, as the reviewer states. However, it translates to such small absolute currents, that it cannot be measured reliably in these devices optimized for high currents. To probe low current density physics, much larger and longer structures (to amplify voltage drop) would be much more suitable to address this problem.

In summary, I had asked the authors to demonstrate a clear advantage or improvement of their experiment over previous studies. Unfortunately, no such advantage has been established in the present version of the manuscript.

Reviewer #2 (Remarks to the Author):

In this resubmission of their work on “Electronic dimensionality of UTe_2 ”, the authors have made their answers to the referees’ remarks and added some new results. Reviewing the answers to the referees:

- On the question of a difference between measurements on bulk samples and FIB microfabricated samples: the authors do explain convincingly that there is in fact no discrepancy regarding the results

on flux-flow, and that their sample quality and geometrical definition are at least as good as on the bulk measurements. Regarding the value of the metamagnetic field, a similar difference from sample to sample is found in both cases. It is more surprising to find such a variation in the FIB microfabricated samples, because they are cut in the very same crystal and probably in very closed area of the sample. This remains one of the puzzles of this compound but indeed, it does not question the quality of FIB shaped samples.

We thank the reviewer for their insightful reading and comments on our work. Indeed one expects a match between the two bars cut from the same sample. We attribute this small deviation to the different directionality of the uniaxial strain due to substrate interaction, that concomitantly yields a small difference in H_{c2} . We have clarified this in the final version, which may also point to strain as an uncontrolled factor explaining the variability of the metamagnetic field and the physical properties at large.

- Regarding the question of the comparison between the resistivity measurements by Eo et al. on c-axis transport on (early) bulk CVT crystals with the author's results, I understand the point that the paper by Eo has long been central in the discussion of the transport anisotropy in UTe₂. However, it is also known presently that these difficult measurements were probably done with no perfect alignment of the current with c-axis and so significantly underestimated the c-axis resistivity. Hence, I find that discussing the more recent bulk results would have been helpful to define the puzzle of the c-axis transport and its non-evolution with sample quality.

We appreciate and resonate with this constructive comment. In the current version we now discuss also two bulk results from both MSF and CVT crystal to develop a more comprehensive case. It would be interesting to revisit the geometries studied by Eo *et al.* given that a far from perfect alignment would be necessary to reconcile their data with other experiments.

- Regarding the interpretation of the large anisotropy observed between a,b electrical transport with c electrical transport in the present measurements, I still find the discussion confusing: in their rebuttal letter, the authors, if I understand well, distinguish a " ", which would be realised in the SC₂ phase of UTe₂ as demonstrated by the lock-in transition they observed in the vortex state, from a "quasi 2D conductor", which would imply, like for the cuprates, that the coherence length along the c-axis is shorter than the interlayer distance.

However, in the presentation of the results in the main paper (lines 98-99), they state from their transport measurement that UTe₂ is a layered conductor "not quite in the quasi 2D-limit". Maybe the problem is to define what is a layered conductor then? However, from the crystal structure or the cleaving properties, it is not easy to picture UTe₂ as layered (along the c-axis), as opposed to organic, cuprates, van der Waals compounds...

We agree with the comment on the wording "layered". While in some cases (cuprates, organics, vdW,...), structural layers define anisotropic hopping terms and thereby naturally set the direction of a 2D Fermi surface, in other cases like UTe₂ this anisotropic hopping is not reflected in the crystal structure so immediately. Therefore we now don't call UTe₂ "layered" in the text anymore and have streamlined the overall discussion of anisotropy. This is a valuable comment as "layeredness", "lock-in effect", "quasi-2D" all encode various playforms of anisotropy, yet are not interchangeable.

An additional confusion comes from the comparison to the Fermi surface (warped cylinders), which the authors claim to be in agreement with their transport measurements: the sentence 114-117 state that: "Our results paint the picture of a much more anisotropic normal state in the MSF crystals, which is further supported by the observation of strong vortex lock-in in the high field phase

discussed below. The large transport anisotropy observed here is well explained by a corrugated cylinder alone, suggesting that UTe₂ is a low-dimensional metal”.

These two sentences suggest again that UTe₂ is a 2D metal, contradicting observations mentioned above on the SC1 phase, or the global conclusion of the paper that anisotropy in the SC2 phase might arise from a modulation of the superconducting state, not from normal state properties. And makes it hard to understand the reason for the suggestion of anisotropic diffusion on line 120, if the c-axis transport is well explained by the sole Fermi surface?

We thank the reviewer for raising these points. In our view the normal state anisotropy agrees with the other observations estimating the anisotropy of SC1 phase. The coherence lengths from STM measurements give that the ratio $\gamma = \frac{\xi_{[01-1]}}{\xi_c}$ is likely in the range of 3 to 4, considering that ξ_b is very likely much larger than ξ_c , the anisotropy between ξ_c and $\xi_{a,b}$ would very likely have a value corresponding to the transport anisotropy $\frac{\rho_c}{\rho_a} \sim 25$ and $\frac{\rho_c}{\rho_b} \sim 50$, which is supposed to be γ_{ca}^2 and γ_{cb}^2 when only considering effective mass anisotropy and assuming an isotropic scattering. These numbers are well compatible with our observation of the vortex lock-in phenomenon, the experimental evidence for its 2D nature. Therefore we would like to argue that the vortex phenomena found in SC2 evidence that the normal state should be as anisotropic as we see, while it still requires a commensurability between ξ_c and interlayer distance that is hard to reach based on our knowledge so far. We have further clarified this in the text.

I would rather agree with this conclusion of anisotropic scattering and with the related discussion in the rebuttal letter, precisely because c-axis transport cannot be explained solely by Fermi velocity anisotropies. Indeed, the author also show (in the SI) that the geometry of the Fermi surface gives an anisotropy between a, b and c axis in the same order as measured by electrical transport. But quantitatively, a factor 3 is missing at the crude level of their calculation, and making more elaborate estimates is beyond reach due to the correlation effects. They also discuss how the coherence length as measured by STM or the anisotropy of H_{c2} in the SC1 phase does not support “layered 2D superconductivity” but rather an anisotropic 3D conductivity. I do not believe that there is a problem with the conclusion of the authors, but rather a problem in the presentation of the discussion: an external reader will be stunned by the factor 50 anisotropy of the transport, and it would be very helpful to distinguish between what everybody agrees on (that UTe₂ is an anisotropic 3D conductor and superconductor in the low-field phase, with a quasi 2D warped Fermi surface leading to the 3D behaviour, with what is (according to the authors) really supporting/explaining/or contradicting the new result on the factor 50 anisotropy between ab and c resistivities.

We thank the reviewer for this comment, and have further laid out the discussions on anisotropy. We would like to stress that warped cylindrical Fermi surface can indeed induce anisotropy as large as 50 without assuming any anisotropic scattering, as long as one could tune the warping of the pockets like Eaton *et al.* did in their discussion of the paper. In that case, they also didn’t count in the *k*-dependence of the $|\mathbf{v}_F|$, which could be another source of anisotropy. Meanwhile, we agree that it is possible that anisotropic scattering is playing a role, therefore we deleted the word “alone” in line 117 of the previous manuscript. It is also true that the coherence length anisotropy only supports an anisotropic 3D conductivity in SC1 phase. But as mentioned above, it doesn’t contradict with the high anisotropy we observe in the normal state.

Regarding also the question of anisotropy of the coherence length, directly proportional to the Fermi velocities, the authors still present, with caution, their estimation from extrapolation of the upper critical field to zero temperature, explaining that in a field-tuned system, they cannot be relied-on. In this new version, they added references to the recent STM work on the vortex lattice. Let us remark

that these STM estimates of the coherence length along the different axes (directly proportional to the average Fermi velocity in that direction) are in very good agreement with previous estimations deduced from H_{c2} measurements, taking account of the field tuning and using the initial slope of the upper critical field (at T_c) rather from the zero-temperature extrapolation (see references 9 and 33).

We fully agree with this statement, however it appears a bit more subtle. As orbital limits are concerned, dH_{c2}/dT measurements at T_c and vortex core imaging via STM should agree, and it is indeed gratifying that it does. However, both are measured in the zero-field limit, and thus reflect the ground-state values in the absence of substantial field tuning. Under high fields, the coherence lengths do evolve away from these values.

- On the main result of the paper, namely the very large anisotropy in the SC2 phase between flux-flow resistance for current along the c or along the a-axis, the authors have clarified several points as explained in their rebuttal letter, and their conclusion appears more clearly.

- On point raised by both referees was the lack of points to discuss the shape of the angular dependence of the lock-in transition. This was corrected in this new version thanks to new measurements with continuous rotation. The discussion is also clear except for the last sentence (lines 224-226): "This increase is reminiscent of the growth of the quadratic temperature coefficient of resistance in a recent Kadowaki-Woods analysis³⁵, suggestive of a fluctuation-driven weakening of the superconducting coherence as the metamagnetic transition is approached." This suggests that superconductivity would become weaker due to the increase of thermal fluctuations? However, such an effect is characteristic of the strong coupling regime, and at the opposite, the increase of the A coefficient or of the Sommerfeld coefficient was taken in the community as showing an increase of the pairing strength, quite contradictory with the picture of coherence weakening?

This is an excellent catch, thank you. It is a typical issue of phrasing something in a too compressed way, losing its meaning on the way. The weakening of vortex confinement implies that the lock-in criterion $d_c < 2\xi_c$ is less fulfilled. This could happen if d_c shrinks in principle, if it was a field-tuned incommensurate modulation, yet this would be quite exotic. Alternatively, ξ_c could grow which lowers the stiffness. This is now corrected.

- Regarding the comparison between the thermodynamic phase diagram and that deduced from flux-flow resistance, the authors are cautious in trying to identify anomalies on transport reflecting the SC1-SC2 transition. Nevertheless, or maybe because of this caution, the statements remain ambiguous: lines 190-193 suggest that the step-like accident at 21.7T could mark the SC1->SC2 transition, but lines 195-200 seem to discuss the irreversibility line observed for example on magnetostriction within the SC2 phase, whereas line 200-201 comes back to the SC1-SC2 transition. This is all the more confusing that the SC1-SC2 line is the only one observed at low temperature (.3K), and that it does decrease in field on increasing temperature, which would be compatible with self-heating effects, whereas indeed the irreversibility line has the opposite behaviour, and was not observed at such a low temperature.

We thank the reviewer for this comment. We now state more clearly that H^* , which is a transport signature, should correspond to irreversibility line rather than SC1-SC2 transition. We also modified line 200-201 to discuss irreversibility line rather than SC1-SC2 transition, although these two things can be closely related. We also appreciate the comment about caution. In our view, this field suffers from strong claims made on rather circumstantial experimental evidence, which is why we note coincidences or correlations between other probes in hopes to contribute to a more complete picture, yet abstain from connecting subtle transport features too strongly with certain microscopic ideas. Naturally, this should bring clarity and not come at the price of confusion. We have combed through the entire manuscript to remove such sources of inconsistencies.

- Last, the introduction of the discussion is still confusing: it seems to start with a claim that UTe₂ is a low dimensional quasi2D superconductor finishing line 254, until this starts to be questioned on line 275. Again, there is no problem with the physics, but it could be easier to follow if it was announced earlier that the most straightforward interpretation raises some difficulties requiring to look for another scenario.

In line 254, we don't directly claim UTe₂ as a quasi-2D superconductor, but rather "suggest to consider UTe₂ as a quasi-2D superconductor" based on its anisotropy. This is to find an explanation of the exotic vortex physics found in SC2, which so far could only be attributed to quasi-2D superconducting state. Also later on line 275, we questioned on how could a quasi-2D superconductor be realized. This follows the less common logic that experimentally we found results that are only compatible with the quasi-2D superconductor model, but meanwhile we are still not knowledgeable enough to claim it as quasi-2D superconductor by showing the defining criteria $\xi_c < d_c/2$ is met. We rephrased some sentences in this part with the hope that it is now clearer to the readers.

As a conclusion, I do recommend publication of this work, which brings new result notably on the nature of the high field superconducting phase. I would just recommend to still try improving readability in clarifying the presentation of the point mentioned above.

We sincerely thank you for taking the time and effort to provide such an extensive and thoughtful report. It truly helped improve this manuscript in the way peer review is supposed to work. Thank you!

Response to Reviewers

Reviewer #1 (Remarks to the Author):

The authors report electrical resistivity measurements on FIB-microstructured UTe_2 samples and discuss the possible observation of a vortex lock-in effect. Based on these observations, they consider the possibility of quasi-two-dimensional superconductivity in this compound. However, I have serious concerns regarding both the data and its interpretation, as detailed below. Their interpretation relies on data that are questionable. Studies on microfabricated samples must first demonstrate consistency with bulk properties; otherwise, it remains unclear whether microfabrication may have altered the intrinsic behavior of the material. Due to these concerns regarding data quality and interpretation, I cannot recommend publication in Nature Communications.

The key data set is presented in Fig. 4b, which shows vortex flow occurring only near the $H // b$ orientation. The authors interpret this as evidence for vortex lock-in and as support for quasi-two-dimensional superconductivity. Their data suggest that superconductivity (SC2 phase) persists up to at least 16 degrees away from the b -axis towards the a -axis. However, according to Z. Wu et al. (PNAS 121 (37) e2403067121 (2024)), superconductivity is suppressed above 8 degrees away from the b -axis towards the a -axis. This significant discrepancy with bulk results raises serious concerns about the reliability of the present measurements or the quality of the samples.

Furthermore, the metamagnetic transition field reported in Fig. 4b is approximately 33 T, which is clearly smaller than the widely reported value of 34.5 T for $H // b$. Since 34.5 T represents the lowest metamagnetic transition field for any field orientation, it is highly questionable that the transition field in their measurements appears even smaller, suggesting additional uncertainties.

We thank the reviewer for their careful inspection of our paper and the critique, which did improve the paper by pushing us to do more measurements and analyses. We have taken them seriously and addressed each of the points in details below.

Importantly, we have never measured nor reported a rotation from “ b -axis towards the a -axis”, instead we rotate from b towards c . The field-angle responses for these symmetry-inequivalent rotations are well known to differ substantially and in fact our sample is in concordance with published literature.

We respectfully rebut the notion of “significant discrepancies with bulk results”, this is simply counterfactual. UTe_2 is well known to be a chemically complex material with various uncontrolled factors impacting microscopic parameters, which in turn lead to a well-known substantial spread of parameters. Taking the paper mentioned by the reviewer [Wu, Z. *et al.*, PNAS **121**, e2403067121 (2024)], we here reproduce Fig. 3 and Fig. 6 from this paper (which in turn are compilations from other works), as well as Fig. 3 of [Sakai, H. *et al.*, *Phys. Rev. Materials* **6**, 073401 (2022)].

REDACTED

In all physical quantities vast variances between different experiments are found, including T_c , field angles and the metamagnetic transition field. Owing to the complexity of the material, this is the unfortunate state of the art in this compound. Furthermore, this is complicated by the fact that it is unclear which of them is a measure of how close the real material is to the idealized UTe_2 , akin to the role of RRR as a measure of crystal quality of a metal. MSF grown samples consistently show higher T_c than CVT, yet there are still debates which one is closer to perfect UTe_2 system. In one line of argument, impurities degrade unconventional superconductivity as in e.g. the case of Sr_2RuO_4 , while superconductivity can also be strongly impacted and enhanced by chemical doping via vacancies, as in the cuprates. In short, as the “intrinsic behavior of the material” is not clarified, a scientific basis is missing for the suggestions of an “alteration of (the intrinsic behavior) by microfabrication”.

In fact, all experimental evidence supports the opposite: the microstructured UTe_2 crystals are in full agreement with published literature on bulk crystals, thus proving the “consistency with bulk results” as requested by the reviewer. First, let us reiterate that these samples yielded the second-ever report of Shubnikov-de Haas oscillations in quantitative agreement with bulk crystal measurements of de Haas-van Alphen oscillations – a universally accepted rigorous testimony of crystalline purity and unchanged electronic spectrum. There are many papers on high-field transport in bulk crystals that did not achieve this.

In the supplement we quantitatively demonstrate that the phase boundaries obtained on our structures are in excellent agreement with the scattered data on bulk crystals, both in field angle θ_{bc} as well as in field H_m (see Table S3,4 and Fig. S5).

Given the observation of SdH oscillations and the clear consistency of our results with bulk measurements, clearly the microstructures are of superb crystalline quality reflecting the intrinsic physics of UTe_2 .

There have been previous reports on vortex flow in UTe_2 . Including the present work, I find three papers addressing flux flow. H. Sakai et al. (PRL 130, 196002 (2023)) reported flux flow across the entire SC2 phase for $J \parallel a$, while Y. Tokiwa et al. (PRB 108, 144502 (2023)) observed flux flow only in the intermediate field region of SC2 for $J \parallel a$. In contrast, the present work claims no flux flow at all for $J \parallel a$. The authors should explicitly discuss the advantages or improvements of their measurements relative to these previous studies. Otherwise, readers are left uncertain as to which result reflects the intrinsic behavior.

Unlike the reviewer suggests, there is no discrepancy between these and our results. On the contrary, it is gratifying to see such consistency.

The study by Sakai *et al.* precisely reproduces our observation. Importantly, they cannot access the temperature range of our experiment. Instead, they observe flux-flow resistance at high temperature ($T = 1.0$ K) close to the phase boundary. In fact, as they cool towards their base temperature of 0.5K, the flux flow resistance goes to zero, in full concordance with our data – and further evidence for the absence of substantial self-heating as discussed later.

At first glance, one might interpret the observed flux flow for $J \parallel a$ in the intermediate field range (14-22 T) by [Tokiwa, Y. *et al.*, *Physical Review B* **108**, 144502 (2023)] at odds with our observation of its absence. Importantly, we detect also a sharp angle dependence of the flux flow in ρ_a as it only vanishes for fields exactly aligned with the b -direction. At small misalignment angles ($\theta_{bc} = 4^\circ$), our Fig.4 shows clear non-zero flux flow for $J \parallel a$ in this intermediate field range which is naturally explained by weakening of intrinsic pinning due to misalignment between field and quasi-2d layers. A natural reconciliation is a small misalignment angle in their experiment, which is common in manually aligned probe sticks without rotation capability. Notably, Tokiwa *et al.* themselves conclude a misalignment of at least 2° w.r.t. the b -axis from the inconsistency of their obtained T_c . Unlike their setup, we have full control over the field angle via the rotator which is necessary to study sharp features exactly at $H \parallel b$. This is not a critique on the work of Tokiwa et al, it is well-known that sharp features such as lock-in states are readily missed by static experiments and can really only be found systematically by field rotation experiments.

One important factor is the precision of the current alignment along the a - or c -axis. Ideally, the distance between voltage contacts should be much larger than the sample's cross-sectional dimensions. It remains unclear whether the sample geometry in the present study is superior to that of the previous work.

With all due respect, this statement is a distortion of the experimental situation. Our microdevices are nearly mathematically perfect rectangular cross-section. The bars are cut after precise XRD alignment from the parent crystal with a precision of better than 0.5° . We show fully quantitative measurements of the geometry and images via scanning electron microscopy, such that the current paths can be quantitatively modeled. The full fabrication and contacting procedure are given in detail in the methods section.

We fully agree with the reviewer that “It remains unclear whether the sample geometry in the present study is superior to that of the previous work.”, simply because other works do not nearly as openly and completely publish their sample information, including the Tokiwa and Sakai papers. In Tokiwa, the entire information is “The rod-shaped sample used for the measurements has a cross-sectional area of $0.29 \times 0.26 = 0.075 \text{ mm}^2$ and a length of 2.2 mm along the a -axis between the

voltage contacts". Sakai shows a rectangular drawing and the statement "A crystal was selected with a size of $0.73 \times 0.75 \times 4.6 \text{ mm}^3$ and RRR 180". That is all the information about the samples given in the two papers. Obviously, current path homogeneity and robustness of results strongly varies on how the contacts were made (contacting method not stated), the contact area, the exact configuration and symmetry of the contacts and how close the real sample shape is to this rectangular approximation. Given that essentially nothing is known about the geometry in other works, it is indeed difficult to assess if ours is "superior", however it is clearly very good.

Importantly, this is not meant as a critique on their work. Their thorough works follow the established state-of-the-art in our field. If anything, it is a critique on the field at large that in the past even published diagrams of cryostats and in-depth information on samples, and nowadays provides sparse sample information at best.

Yet in all fairness one must admit that the high level of detail we provide on all samples leaves nothing "unclear" about geometry of the samples we studied. We have further expanded on this via an in-depth discussion of all sample geometries in the main text and added more extensive sample descriptions to the supplement.

I also find the fitting results presented in Fig. 3b unreliable. The data are extremely sparse for fitting with any function: there are not even two data points within the half-maximum width. Furthermore, the authors do not present data for negative field angles, making it impossible to verify that "0°" truly corresponds to $H // b$. If the actual $H // b$ orientation is slightly offset from 0° , the fitting results would change substantially.

Thank you for raising this point. Indeed these datapoints were sparse, due to issues with the angle readout on that particular run we decided to slice the phase space via field-scans at constant angle. Zero degrees, defined by the peak maximum, was searched for carefully manually, thus an offset from 0° distorting the width was not a concern to us. Still, this is not ideal and the fitting is not the best.

We take this feedback also raised by reviewer 2, and performed additional high-field measurements using continuous rotation, which are now discussed in the manuscript. Indeed, the Lorentzian shape is a crude approximation for the real shape of the peak. The main result of a very narrow peak, however, is robustly reproduced in this experiment. Importantly, now the maximum is resolved well eliminating remaining issues of peak alignment. We now changed the width discussion to the full-width half-maximum of the real shape as measured. Despite the crude shape approximation, the mathematical FWHM of the Lorentzian fit is very close to these new measurements, thus confirming the previous statements.

It would be interesting to draw more quantitative information from the shape, however we are not aware of theoretical work modeling the angle-dependent peak shape in a systematic manner. It is evidently an interplay of the geometry, the surface and bulk pinning, the anisotropic coherence length and penetration depth as well as the applied current. Still, we fully concur that eliminating this residual doubt was a necessary step in this project.

The authors assign the H^* anomaly to the phase boundary between SC1 and SC2. However, it is difficult to distinguish this assignment from an accidental coincidence. The authors need to present the temperature dependence of $H^*(T)$ and demonstrate that it consistently tracks the SC1–SC2 phase boundary to justify this interpretation.

We agree with the reviewer that this feature is an intriguing aspect of the data and that it is worthwhile exploring in more depth. Currently, we do not have this data for technical reasons. First, all experiments were performed at base temperature submerged in ^3He to eliminate the effects of self-heating (see next comment). Second, with the inability of the magnet system to go to zero field to read out thermometers without magnetoresistance errors renders a fine-grained temperature dependence in this range quite challenging.

Although this field value matches the irreversibility line, it is difficult to rigorously assign signatures in transport with a phase transition between two superconducting phase. Hence we did not define this critical field as the SC1-SC2 boundary, but rather pointed out its proximity to the previous reported value of that phase boundary ($H \approx 19$ T). We have further softened this tangent point, and merely note the correlation, which is self-evidently true.

Finally, in Fig. 2c, the authors report flux flow even when the DC current is zero. I am concerned that this flux flow may be induced by the finite AC current, and it is necessary to verify whether the voltage drops in the limit of zero AC current.

We apologize that we do not understand the meaning of this comment or critique. Naturally the finite voltage deep in the superconducting state at zero dc bias current is the response of the very weakly pinned vortex lattice to the ac-current, as common for systems in the lock-in configuration. Thus we do not see what aspect of this natural fact is the cause for concern? As for the second part, it is obvious that the ac-voltage drops to zero in any system under zero ac-current? We regret if we likely did not respond here to the point, we honestly did not capture its meaning.

The authors should also examine the possible effect of Joule heating. Nearly ohmic behavior is observed around 18–20 T, which coincides with the field region where $T_c(B)$ is lowest. This near-ohmic behavior could result from Joule heating elevating T_{sample} above T_c specifically in this field range, or simply from a reduced critical current because the measurement temperature becomes closest to T_c at these fields.

Self-heating must clearly be discussed in the manuscript, we thank the reviewer for pointing this out. Naturally we had performed self-heating checks, which we had missed to state in the manuscript. Now, there is an extended discussion in the supplement.

As shown in the supplement, we performed IV measurements at higher field angles in the paramagnetic phase and detected measurable self-heating at 15 kA/cm². This region is far away from either the SC phases or FP state in the phase diagram thus excluding any other potential origin of nonlinearity. As we focus on the onset of vortex motion, we limit the currents in the paper well below 0.5 kA/cm², thus safely within the linear regime. This is not surprising given the low power applied to the sample and its high conductivity. The maximal current applied was $I = 50 \mu\text{A}$ to the sample of typical 2-point resistance of $R = 2 \Omega$, thus a power of $P = RI^2 = 5 \text{ nW}$. In light of the ohmic, large area gold contacts (see methods) and it being submerged under liquid, this is negligible as observed.

Fig. S6. Nonlinear characteristics of sample S3 measured at $H = 44 \text{ T}$, $\theta = 55.7^\circ$, $T = 0.3 \text{ K}$. **a.** J - E curve, with dashed red line as a guide to the eye for linear (ohmic) behavior. **b.** ρ - E curve, with dashed red line as a guide to the eye for constant resistivity.

We again thank the referee for raising this point and stimulating this important addition to the paper.

Reviewer #2 (Remarks to the Author):

Report on the paper “ Electronic dimensionality of UTe2 “ by Zhang et al.

In this work the authors report experimental measurements of the non-linear electrical transport for fields applied around the b-axis in the different superconducting phases of UTe2 in FIB microstructured high-quality (molten flux grown) single crystals. This microstructuration allows to reach very large current densities usually only possible in thin films, and hence to directly study the different flux-flow, flux-creep regimes across the superconducting phase diagram. Moreover, the microstructuration also permits to study the transport for different current directions on the same crystal, which proves to be highly pertinent, notably in the superconducting phase.

However, a first surprising result is the large resistivity observed on high quality samples for current along the c-axis:

We are delighted to receive this comment and thank you for the expressed interest in the matter. In fact, we had a discussion about the emphasis to put on the discussion of the normal state transport, as it may appear a bit niche. However we feel this is a most important observation, and we appreciate the opportunity to expand it in the revised version.

it was known from measurements on millimeter-size samples that resistivity is larger for current along c than for current along a and b .

While we fully agree with this experimental situation, it is fair to mention that the work by Eo *et al.* is prevalent in discussions in the field, and the apparent isotropy remains an argument for the elusive 3D pocket. We specifically discuss this reference, as it is in direct contradiction when it comes to the levels of anisotropy – which is key to the main aspect of intrinsic vortex pinning.

However, the anisotropy between the c -axis and the other directions was more than 5 times smaller. Everything happens as if the improvement of the RRR on the molten flux sample studied here did not affect the c -axis direction: the values observed here are comparable to those on works a bit more recent than that of Eo *et al.*, performed on CVT samples when the current direction is indeed along the c -axis (see PRB 109, 155103 (2024)).

Otherwise, the value of the c -axis conductivity can be significantly smaller (see again PRB 109, 155103 (2024)). Hence, rather than a difference in the anisotropy between CVT-grown and flux grown samples, due to the “non-trivial role of chemistry”, don’t these results rather suggest the existence of defects or impurities affecting transport along c -axis more strongly than the sample stoichiometry improved by the flux-growth process?

We completely agree with this statement, yet we fail to see the contradiction between “non-trivial role of chemistry” and “the existence of defects or impurities affecting transport along c -axis more strongly than the sample stoichiometry improved by the flux-growth process”. That is exactly the point, different growth processes appear to impact the effective transport anisotropy. From a purist perspective, a warped cylinder and a relaxation time approximation yields Boltzmann results that acceptably describe anisotropic conductors. Here, one needs to introduce a tensorial form an anisotropic scattering term $V(k, k')$ that modifies the in- and out-of-plane lifetime separately. Simple point defects are not effective at this, so somehow more directional scattering centers form under one growth condition compared to the other. This is exactly what we meant by “non-trivial role of chemistry”, which we now state in the revision.

At the same time, the microscopic insights into defects and local chemistry are very limited from transport, and it appears not warranted to go too much into depth of the origin – which was the reasoning for the deliberately vague use of “non-trivial role of chemistry”. There is a lot we still do not know about the physical chemistry of UTe_2 . Given the electronic system is strongly renormalized by Kondo physics, it is for example not inconceivable that differences in the effective anisotropy come from orbital selectivity of Kondo physics, which may in turn yield anisotropic Kondo scattering. The role of the U-dimers has been established as key in this compound, and Te vacancies/interstitials may substantially distort the crystal fields at dimers favoring some direction over the other. As we could only speculate, we hope you find the current careful phrasing in the paper adequate to reflect the experimental observations we present.

So, from the observed very large anisotropy of the electrical transport between the c -axis and the a - b direction in these high quality microstructured samples, the authors conclude that UTe_2 would be a quasi 2D layered conductor, comparable to the High-Tc cuprates, iron-pnictides or organic systems. This point is indeed a recurrent question in the analysis of the physical properties of UTe_2 , and the new results presented here on microstructured samples which are both “high quality samples and geometrically well-defined is an important element.

The conclusion is however maybe still not so well firmly established, because if the point was only that of a quasi-2D anisotropy, resistivities should scale in all direction with the RRR.

Indeed, yet only under the condition that the electronic spectrum itself is temperature-independent, which would allow an interpretation of RRR as the effective ratio of disorder scattering over phonon/thermal excitations. Here, the electronic system itself is strongly temperature dependent due to Kondo coherence. The Fermi surface that describes UTe₂ at 2 K is clearly different from the much more isotropic one at high temperature. It is no surprise that DMFT picks up this strong temperature dependence of the Fermi surface, as reported in [Halloran, T. *et al.*, *npj Quantum Mater.* **10**, 2 (2025)]. Whether DMFT captures the correct Fermi surface is a different story of course, especially given the isotropic treatment of the fluctuations and its inability to access the relevant low energy scales for 2 K physics. Still it appears fair to say that the electronic system itself is substantially changed by temperature, and with it the relative ratio of directional Fermi velocities, that the interpretation of direction-dependent RRR values is difficult.

Moreover, as remarked by the authors, this very strong anisotropy of the Fermi velocities would be expected to reflect in the superconducting properties, and notably in the anisotropy of H_{c2}: the author do show that even with the uncertainties arising from the “field-tuned” superconducting properties, and the departure of the temperature dependence of H_{c2} from the conventional WHH predictions, the coherence length along the c-axis far exceeds a putative “interlayer distance”, contradicting the hypothesis of a layered superconductor.

Thank you for raising this point, which we hope to have clarified in the revision. These statements are all correct, and consistently they apply to the SC1 phase in which no lock-in effect was observed. In this state, the coherence length by all methods of estimation exceeds the c-axis unit cell greatly, leaving it an anisotropic yet still 3D coherent superconductor. In SC2, the lock-in effect provides direct evidence for such a layered superconducting texture.

Yet, the core of the present work is certainly the study of the “flux-flow” state in UTe₂ for field along the b-axis, both in the low field SC1 phase and in the SC2 phase up to the metamagnetic transition at around 35 T. Several striking results are presented:

- transport along a-axis displays very little flux-flow voltage, showing that vortices remain strongly pinned for current along the a-axis (and field along the b-axis) both in the SC1 and SC2 phases.
- transport along the c-axis displays sizeable “flux-flow” voltage at low temperature above 15 T, and a marked anomaly at the transition between the SC1 and SC2 phases (between 21-22T).

The huge difference between a and c-axis transport is a very new result, and add significant input compared to previous work (ac susceptibility and magnetostriction) which did show the existence of weak pinning areas in the superconducting phase diagram for H//b. There is a good qualitative agreement between the different measurements, and the present work could also study the angular dependence of the flux flow resistivity which typically vanishes above 4° misalignment in the (b,c) plane.

If there is little doubt on this sensitivity to angle, the attempt to fit the angular dependence by Lorentzian forms is surprising, owing to the sparse number of points notably in the region of the line width. Hence, trying to extract quantitative information on the line width as present in Fig 3.c is not convincing and maybe not so important either for the discussion.

Thank you for your assessments on the novelty of our work. Indeed we acknowledge and resonate with the description of the shortcomings of the Lorentzian fit presented. This happened because of the difficulties of measuring a signal during rotation on that particular magnet run. Given the limited access to fields of that magnitude, we were not able to measure at tighter angles.

We take this feedback, and performed additional high-field measurements using continuous rotation, which are now discussed in the manuscript. Indeed, the Lorentzian shape is a crude approximation for the real shape of the peak. The main result of a very narrow peak, however, is robustly

reproduced in this experiment. We now changed the width discussion to the full-width half-maximum of the real shape as measured, complementing the larger dataset of fixed-angle scans. Despite the crude shape approximation, the mathematical FWHM of the Lorentzian fit is very close to these new measurements, thus confirming the previous statements.

It would be interesting to draw more quantitative information from the shape, however we are not aware of theoretical work modeling the angle-dependent peak shape in a systematic manner. It is evidently an interplay of the geometry, the surface and bulk pinning, the anisotropic coherence length and penetration depth as well as the applied current. Still, we fully concur that eliminating this residual doubt was a necessary step in this project.

Indeed, the main point is clearly the origin of the anisotropy between flux-flow along a and along the b -axis. The authors discuss and exclude a lock-in of the vortices between superconducting planes for the reasons on the coherence length along the c -axis mentioned above. This is perfectly sound, however, the estimation of the lower bound of $H_{c2} = 145$ T in the SC2 phase is difficult to understand: as mentioned before, UTe2 is a “field tuned system” hence it is complicated to understand how to extrapolate H_{c2}/b in the SC2 phase from coherence length in the SC1 phase which depend on the T_{sc} value in that phase. More deeply, what is the meaning of extrapolating H_{c2} in a field-tuned system, and what for?

Thank you for this comment. Upon reading this section again in light of this comment and discussions with colleagues, we realize that we did not properly communicate the main point clearly – causing this understandable confusion. The main point of the completely inadequate coherence length analysis via the upper critical fields was to show its failure. It yields unphysical results. $H_{c2} = 145$ T is not an extrapolation though, but rather an equivalent way to discuss the Ginzburg-Landau coherence length at a particular field value. In field tuned systems, as you say, it has no predictive power, its simply a parameter.

We apologize for this confusing discussion, which was substantially changed in the revision. Instead of building up this analysis to show its failure, we now argue directly with the beautiful experimental data on vortex core STM spectroscopy, which directly mapped the actual values of SC1 coherence lengths. In the meantime, these results have been beautifully confirmed by 2 other groups, a rare case of consistency in this field. Then, we just briefly and explicitly mention the shortcomings of the upper critical field analysis, substantiating the fact that the limiting fields observed in experiment, H_{c2} , have nothing to do with orbital limits but rather with field-tuning of the SC state itself. We hope the revision now makes explicit how inapt this analysis is and avoids quoting absurdly wrong numbers such as $H_{c2} = 145$ T.

The second proposal in the discussion is a modulated phase in the superconducting state (“pair-density wave” state). Such a state could indeed relax the constraint on the coherence length along the c -axis having to be smaller than in the interlayer distance. However, the author should clarify why they expect a change of anisotropy between the SC1 and SC2 phases: in their measurement, flux-flow voltage appears also in the SC1 phase and again only on the c -axis transport.

Within SC1, we do not observe substantial flux flow or lock-in effects – all data points to a more isotropic state here. The impression of flux flow appearing only for $J \parallel c$ may be distorted by our use of a logarithmic color scale. Our intention here was to present the data in all details, maybe this then visually overemphasizes the exponentially suppressed “foot” of the lock-in transition too much.

First, entering into a lock-in state through a field-tuned SC-SC transition is an entirely novel situation for which no analogous observation or theory exists. There is a framework of thermal transitions though. The coherence length diverges at T_c , hence slightly below T_c any superconductor is in a 3D

limit. As the system is cooled, the temperature-dependent coherence length shrinks and eventually $\xi_c(T)$ falls below the interlayer distance, leading to a true Josephson-vortex case. A small excess voltage for in-plane field alignment is observed even in the 3D state at high temperatures, as the Abrikosov vortices start to sense the interlayer modulation of the superfluid density. This phenomenon is highly reminiscent of this residual flux flow voltage in the lock-in configuration within SC1.

Still, we reiterate the complete lack of theory for such a transition, and the analogy between temperature- and field-driven lock-in transitions may be substantially distinct. As such, we also do not have any basis for expectations for the anisotropy – we simply report the observation that the anisotropy changes by a factor of at least 5-8 (in the case no PDW is formed).

Moreover, thermodynamic measurements of H_{c2} have consistently shown that the initial slope of H_{c2} at T_{sc} along the a-axis is larger than along the c-axis: field-tuning prevents again clear-cut conclusions, but the bare measurements do suggest that coherence length along the c-axis is also the shortest in the SC1 phase, which is in line with the present measurements?

Yes. This again is a result of the confusing previous presentation. The whole point was, exactly, to show how determinations of $H_{c2}(0K)$ fail to make statements on the coherence length. Naturally, ξ_c is the shortest length given the layered conductor; the fact it did not come out as such was supposed to corroborate the failure of the analysis. This has been removed and replaced by the STM data, which provides the actual values at $T = 0$ K (a bit less disorder-influenced than dH_{c2}/dT extrapolations, but clearly consistent therewith).

A last point which should be clarified in the discussion is that on the possible Lebed mechanism. Again, this Lebed mechanism seems only possible for reasonable field values only in strongly layered systems, excluded by the authors. Also, the observation of a halo of superconductivity for the field-reentrant phase above H_m seems incompatible with a Lebed mechanism requiring alignment of the field within the layers. Again, this point is confusing and does not seem key for the work.

Thank you for this point. Indeed. The Halo is an interesting observation, yet the question of its relevance to SC2 is not solved. SC2 and the Lazarus-phase (SC3) may be the same state or fundamentally different. They are separated by a strong first-order transition on the U-state, which in a way completely reshuffles the microscopic physics. It is well possible that a Lebed-like mechanism explains SC2 in the vicinity of $H \parallel b$, and an entirely different mechanism yields SC3. The observation of the orphan state in SC3 may be taken as support for such a scenario. We agree it is not central to the observation of an unexpected physical phenomenon, and have toned its discussion down accordingly.

As a conclusion, I do recommend publication of this work in Nature communication, as it bring new important results on the behaviour of the superconducting phases, with unique measurements on high quality crystals, and suggests an appealing hypothesis of a modulated phase at high-field along the b-axis. As mentioned above, some points should be clarified in the discussion and some additional minor points could also improve the manuscript:

- on line 222, add references for the previous estimates of the coherence length in the a and c directions.

Done, this is now given via the STM experiments.

- If available, a complete I-V curve at some field and temperatures would also help distinguish flux-

flow- flux creep regimes and appreciate the values of the critical current

- Measurement of H_{c2} in the different directions on this sample would also be helpful

We fully agree, yet this will require an additional high-field run unfortunately. We do present now the IV curve taken in the paramagnetic state as part of the initial self-heating checks.

- On figures 2, 3 and 4, it appears that H_m is slightly above 35T on the transport measurements with current along a-axis, and around 33.5T (?) for current along c-axis.

Could the authors comment/explain this discrepancy?

As noticed also by reviewer 1, there is a difference in H_m between the devices. There is a substantial variance of this transition field value in literature, and our experiments fall right into the scatter as is shown now in the supplement. It is likely that this now only becomes evident as we directly juxtapose two samples and focus on their differences.

We do not have a microscopic understanding of the relevant factors for this variance, arguably it is again the “non-trivial role of chemistry”. We can only say that within the structured devices we have no evidence or even hint at a trend that microfabrication impacts its value, it rather appears that one cuts a microscopic amount of material with its own set of chemical defects which in turn defined H_m found.

Again, we thank you for the insightful comments and your time reading our manuscript.